# Structural remodeling of target-SNARE protein complexes by NSF enables synaptic transmission

K. Ian White [1,2,3,4,5] ✉, Yousuf A. Khan [1,2,3,4,5], Kangqiang Qiu[6], Ashwin Balaji [7], Sergio Couoh-Cardel [1,2,3,4,5], Luis Esquivies[1,2,3,4,5], Richard A. Pfuetzner[1,2,3,4,5], Jiajie Diao [6] & Axel T. Brunger [1,2,3,4,5] ✉

Synaptic vesicles containing neurotransmitters fuse with the plasma membrane upon the arrival of an action potential at the active zone. Multiple proteins organize trans-SNARE complex assembly and priming, leading to fusion. One target membrane SNARE, syntaxin, forms nanodomains at the active zone, and another, SNAP-25, enters non-fusogenic complexes with it. Here, we reveal mechanistic details of AAA+ protein NSF (N-ethylmaleimide sensitive factor) and SNAP (soluble NSF attachment protein) action before fusion. We show that syntaxin clusters are conserved, that NSF colocalizes with them, and characterize SNARE populations that may exist within or near them using cryo-EM. Supercomplexes of NSF, α-SNAP, and either a syntaxin tetramer or one of two binary complexes of syntaxin−SNAP-25 reveal atomic details of SNARE processing and show how sequential ATP hydrolysis drives disassembly. These results suggest a functional role for syntaxin clusters as reservoirs and a corresponding role for NSF in syntaxin liberation and SNARE protein quality control preceding fusion.

The membrane-anchored SNARE (soluble N-ethylmaleimide-sensitive factor acceptor protein receptor) proteins provide the energy to drive membrane fusion in various cellular contexts, such as intracellular trafficking and exocytosis. One critical exocytic process is synaptic transmission, in which a synaptic vesicle fuses with the presynaptic plasma membrane in the active zone of a neuron to release neurotransmitter molecules into the synaptic cleft in response to the arrival of $Ca^{2+}$ in the presynapse. While many proteins precisely act in concert to ensure synchronized and efficient $Ca^{2+}$-triggered fusion[1–5], this process is fundamentally mediated by the "zippering" of four SNARE motifs typically contributed by three SNARE proteins in the neuronal context−syntaxin-1a and SNAP-25 on the target plasma membrane (t-SNAREs), and synaptobrevin-2 on the vesicle (v-SNARE)−which come together to form a highly stable parallel four-helix bundle during $Ca^{2+}$-triggered membrane fusion. This cis-SNARE complex is then disassembled in an ATP-dependent manner by the AAA+ protein NSF in concert with the adapter SNAP proteins[6,7].

While the molecular mechanisms of this core fusion cycle and the accessory proteins required to ensure $Ca^{2+}$-dependent neurotransmitter release have been extensively studied[4], the organization of the various components in the crowded cellular milieu remains uncertain. Numerous studies suggest nanoscale organization within the presynaptic active zone, where proteins and lipids[8–11] such as cholesterol and $PIP_2$ colocalize or even phase separate into discrete clusters with implications for organization within the active zone and across the synapse[12].

[1]Department of Molecular and Cellular Physiology, Stanford University, Stanford, CA, USA. [2]Department of Neurology and Neurological Sciences, Stanford University, Stanford, CA, USA. [3]Department of Structural Biology, Stanford University, Stanford, CA, USA. [4]Department of Photon Science, Stanford University, Stanford, CA, USA. [5]Howard Hughes Medical Institute, Stanford University, Stanford, CA, USA. [6]Department of Cancer Biology, University of Cincinnati College of Medicine, Cincinnati, OH, USA. [7]Department of Chemistry, Stanford University, Stanford, CA, USA. ✉e-mail: kiwhite@stanford.edu; brunger@stanford.edu

The SNARE protein syntaxin shows such an organization[11,13]. Syntaxin is conserved from yeast to mammals[14,15]; it comprises a short C-terminal transmembrane domain, a SNARE domain, and an N-terminal Habc domain[16,17]. In the neuronal context, the majority (~67%) of syntaxin molecules form nanodomains of non-uniform density[13,18], often near the active zone[8,13,18–21], and these nanodomains frequently overlap with other essential active zone proteins and even docked granules[11,19,20,22–25]. Morphologically, they are dense, around 50–60 nm in diameter, and harbor as many as 70 syntaxin molecules each[13,18,19,22,26]. Optogenetically-induced presynaptic activity drives an increase in syntaxin mobility[27]. These clusters depend on the syntaxin SNARE domain (H3), which drives oligomerization[28].

Despite extensive study, the functional purpose of these clusters has remained unclear. Moreover, NSF, α-SNAP, and syntaxin appear to interact in vitro[29], and α-SNAP associates with liposomes presenting reconstituted full-length syntaxin[30]. Binary (syntaxin–SNAP-25) complexes of different stoichiometries may also form[31–37], and NSF disassembles these complexes in vitro[31,38–40]. Furthermore, continuous ATP hydrolysis by NSF on plasma-membrane mimicking vesicles with reconstituted syntaxin and SNAP-25 produces a primed state for the most efficient and synchronous synaptic vesicle fusion[3]. These observations support a critical role for NSF not only in recycling SNARE proteins following fusion, but also in regulating the pool of fusogenic syntaxin available to enter the vesicle fusion cycle.

Here, we show that spatially organized nanodomains of syntaxin may serve as conserved molecular protein stores, where NSF likely disassembles syntaxin homo- and hetero-oligomers. Thus, NSF may act as a quality control factor to disassemble off-pathway binary complexes of syntaxin and SNAP-25–a prerequisite for synaptic vesicle priming and fusion. We also describe the molecular disassembly mechanism and present a sequential ATP hydrolysis model by which NSF couples γ-phosphate release to SNARE substrate remodeling, with broad implications for the AAA+ protein family.

## Results

### Syntaxin nanodomains are conserved hotspots of NSF activity

First, we hypothesized that NSF is enriched at syntaxin clusters in the active zone, where it might act to liberate fusogenic syntaxin molecules. To test this, we monitored the colocalization of syntaxin and NSF in PC12 cells, a model system for neuronal secretion, using structured illumination microscopy (SIM). All cells were transfected with syntaxin-GFP and NSF with either a short C-terminal linker followed by mScarlet-I (NSF-mScarlet-I; Fig. 1A), a mutant form of NSF (NSF(ΔN)-mScarlet-I; Fig. 1B), mScarlet-I alone (Fig. 1C), or several catalytically impaired mutants (Supplementary Fig. 1A). NSF-mScarlet-I is hexameric and, together with α-SNAP, retains the capacity to bind and disassemble the ternary SNARE complex (Supplementary Fig. 2A). Clear, punctate co-clustering between NSF and syntaxin is visually apparent upon inspection of the micrographs (Fig. 1A–C); Pearson's colocalization coefficient (PCC) was calculated between the green and red channels to quantitate this (Fig. 1D, Supplementary Fig. 1B, "Methods"). Cells transfected with both syntaxin-GFP and NSF-mScarlet-I showed significant and reproducible syntaxin-GFP–NSF-mScarlet-I colocalization (Fig. 1A, PCC = 60% ± 2%; $p < 0.001$ vs. mScarlet-I only), suggesting an interaction between NSF and syntaxin molecules within the nanodomains. As a control, mScarlet-I alone did not colocalize with syntaxin, showing a nearly three-fold reduction in colocalization (Fig. 1C; PCC = 22% ± 2%). To verify that colocalization is consistent with the binding mode of NSF, in which NSF N-domains engage multiple α-SNAPs, which in turn coat the SNARE complex[6,7,40,41], we examined colocalization of GFP-syntaxin with a mutant form of NSF with the N-domain required for α-SNAP binding–and thus SNARE complex binding–removed (NSF(ΔN)-mScarlet-I)[42]. As expected, this reduced colocalization to a level nearly indistinguishable from the mScarlet-I control (Fig. 1B, PCC = 28% ± 3%; $p = 0.1776$ vs. mScarlet-I only). Two

previously characterized[43] point mutations to the D2 domain also disrupted colocalization between syntaxin and NSF (Supplementary Fig. 1A, B). These colocalization studies suggest that NSF is associated with some clustered, multimeric species involving syntaxin molecules in vivo, and that the interaction is dependent upon the recognition of a left-handed, four-helix SNARE complex. While these results showed clustering of syntaxin in the context of synaptic transmission, we found that clustering also occurs in the case of the conserved syntaxin homolog in yeast, Sso1p, as well (Fig. 1F, Supplementary Fig. 1C). During cell division, the mother cell transfers plasma membrane and cargo to the budding daughter cell via secretory vesicles, and tethering and fusion take place across the daughter cell plasma membrane[44]; we observe Sso1p clusters enriched throughout the budding cell in areas likely undergoing active fusion.

### Discovery of a syntaxin tetramer complex with NSF–α-SNAP

To gain insight into the contents of the syntaxin nanodomains as well as the molecular basis of the interaction between NSF and syntaxin, we purified the complex consisting of NSF, α-SNAP, and soluble syntaxin (residues 2–253, transmembrane region omitted) by mixing the individual purified proteins and performing size exclusion chromatography (SEC), Supplementary Fig. 3A. We note that soluble syntaxin exists as a mixture of monomeric and oligomeric species[36]. We then performed cryo-EM single-particle analysis (SPA) of this sample. Following 3D classification, 5% of the particles formed a class with an overall resolution of 3.75 Å, revealing a complex (referred to as sx20S) between hexameric NSF, four α-SNAP molecules, and a parallel four-helix bundle of syntaxin H3 SNARE domains (sx20S; Fig. 2A, Supplementary Fig. 4, Supplementary Tables 1, 2, Supplementary Discussion). The remaining particles showed no evidence of SNARE substrate. The finding of a syntaxin tetramer is consistent with crosslinking mass spectrometry analysis of purified soluble syntaxin[36], and corroborates the specificity of the system for parallel, left-handed, four-helix SNARE bundles as no other species identified by mass spectrometry were found in complex with NSF and α-SNAP. The sx20S complex thus further supports the notion that NSF and α-SNAP recognize parallel, four-helix SNARE complexes of variable composition[6,7].

Similar to structures of hexameric NSF and α-SNAP bound to the neuronal ternary SNARE complex (referred to as 20S complex) under non-hydrolyzing conditions[6,7,41], the sx20S complex forms a three-tiered architecture (Fig. 2A). NSF comprises three structured domains connected by flexible linkers–the adapter N-domain followed by the D1 and D2 AAA+ domains. Six NSF protomers come together to form the active form of the enzyme, a homo-hexamer. In the top tier, or spire, the NSF hexamer's multiple N domains bind four α-SNAP molecules and a tetramer of syntaxin H3 SNARE domains. This spire structure positions the N-terminal end of the syntaxin tetramer near the pore of the hexameric D1 ATPase ring of NSF; this middle tier is responsible for ATP hydrolysis and SNARE complex disassembly. The bottom tier consists of the non-hydrolytic NSF D2 ATPase ring responsible for the oligomerization of the complex. We note weak, poorly defined density on the outside of the D2 pore, suggestive of a highly mobile Habc domain; this is consistent with observations of the Habc domain in the yeast 20S complex of Sec17, Sec18, and the yeast SNARE complex (including the syntaxin-1a homolog Sso1p), in which the Sso1p Habc domain is found outside of the D2 pore, packed against the surface of the ring[45]. Presumably, the remaining Habc domains extend outwards from the space between the spire and the D1 ring, although they are not resolved in our structures of the neuronal 20S complex.

The spire structure appears dynamic and rocks about relative to the D1 pore, but three of the six N-domains are relatively well-ordered and engaged with the α-SNAP–syntaxin sub-complex. The D1 ring is asymmetric and split; protomers A–F are defined from the bottom of the split to the top. All six D1 domains are ordered and engage the

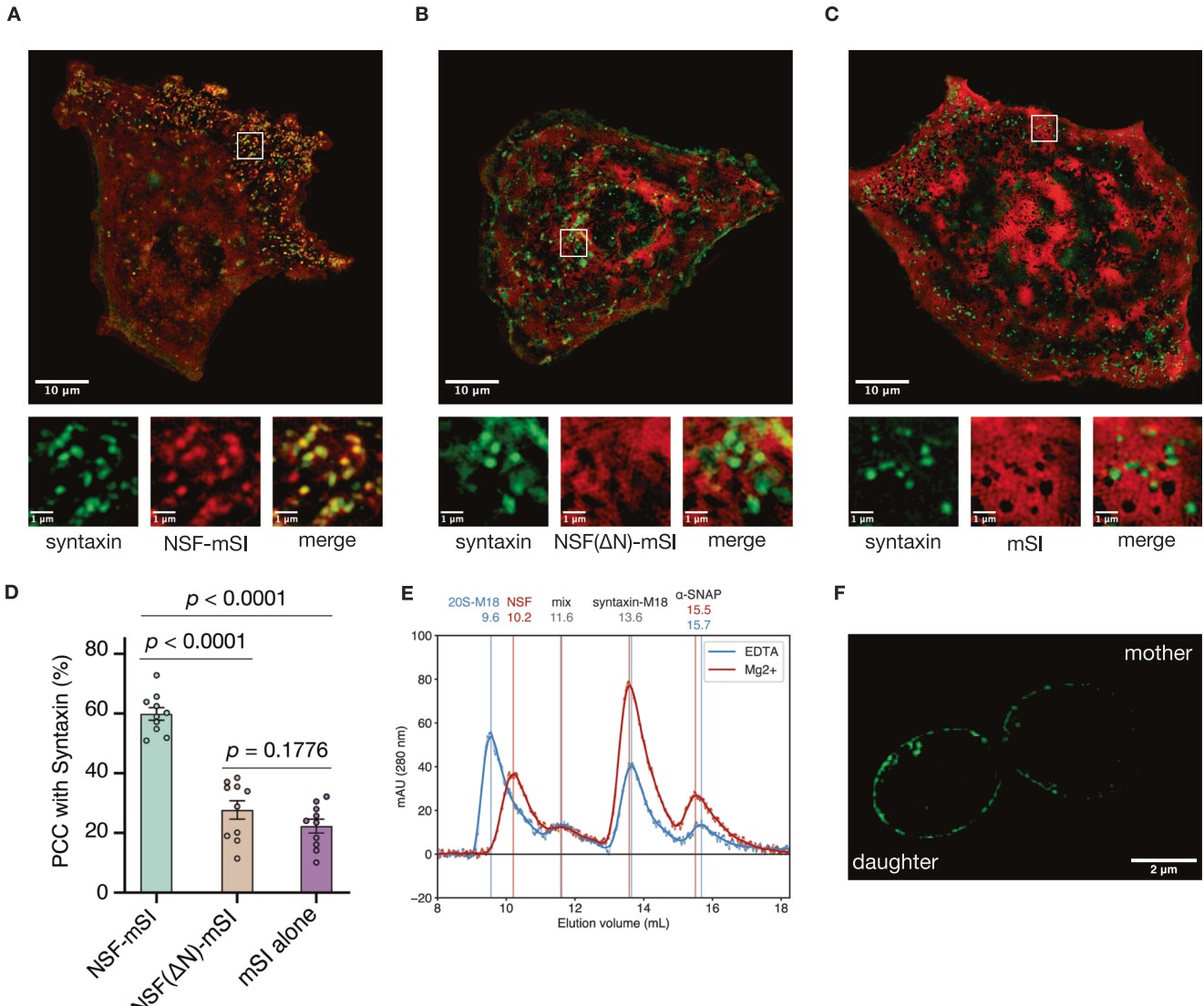

**Fig. 1 | NSF colocalizes to syntaxin nanodomains in an N-domain dependent manner and disassembles syntaxin oligomers. A** Colocalization between syntaxin-GFP nanodomains and NSF-mScarlet-I. NSF often forms clusters overlapping with syntaxin (PCC = 60% ± 2%, mean ± SEM). **B** Colocalization between syntaxin-GFP nanodomains and NSF(ΔN)-mScarlet-I; disruption of colocalization is evident (PCC = 28% ± 3%, mean ± SEM). **C** Colocalization between syntaxin-GFP and mScarlet-I alone. No interaction is apparent (PCC = 22% ± 2%, mean ± SEM). **D** Comparison of NSF-mScarlet-I, NSF(ΔN)-mScarlet-I, and mScarlet-I clustering with syntaxin-GFP nanodomains (M ± SEM, $n$ = 10 images). Statistical significance was established by two-tailed Student's $t$-tests. Colocalization between NSF-mScarlet-I and syntaxin-GFP is significantly different than between NSF(ΔN)-mScarlet-I and syntaxin-GFP ($p$ = 8.9002 × 10$^{-8}$) or between mScarlet-I and syntaxin-GFP ($p$ = 5.2106 × 10$^{-10}$). Colocalization between NSF(ΔN)-mScarlet-I and syntaxin-GFP and colocalization between mScarlet-I and syntaxin-GFP were not significantly different ($p$ = 0.1776). **E** To demonstrate disassembly tetrameric syntaxin-1a SNARE complexes by NSF and α-SNAP, and to show the insufficiency of Munc18 alone in

disassembly, sx20S was prepared from purified components, split into two pools, resuspended in with a ~20-fold excess of Munc18. Following addition of EDTA or Mg$^{2+}$, pools were subjected to size-exclusion chromatography (SEC). Raw and filtered SEC profiles of the non-hydrolyzing (EDTA, blue) and hydrolyzing (Mg$^{2+}$, red) chromatograms at 290 nm are shown. The first peak corresponds to a complex of sx20S-Munc18 under non-hydrolyzing conditions (9.6 mL), or free NSF (10.2 mL) under hydrolyzing conditions. The second peak is consistent with an intermediate species of all proteins (11.6 mL). The third peak corresponds to a complex of syntaxin−Munc18 (13.6 mL), enriched twofold as assessed by integrated area under the curve (IAUC) under hydrolyzing conditions (12.8–14.8 mL; EDTA, 537.5 mAU$^2$; Mg$^{2+}$, 1058.5 mAU$^2$). The fourth peak is dominated by free α-SNAP, enriched twofold under hydrolyzing conditions (14.8–16.8 mL; EDTA, 224.4 mAU$^2$; Mg$^{2+}$, 442.7 mAU$^2$). **F**. Sso1p forms dense clusters in the *S. cerevisiae* plasma membrane during cell division as assessed by SIM. Sso1p clusters were enriched in the budding daughter cell (left); discrete clusters typically approached sizes near the Nyquist limit of 160 nm ($n$ = 46 cell images).

N-terminal linker preceding the H3 domain of one of the four syntaxin molecules via conserved pore loops, which position Y294 such that it intercalates between every other syntaxin substrate side chain passing through the pore (Fig. 2B). The nucleotide states of the NSF ATPase domains are consistent with a pre-hydrolysis conformation; D1 protomer A is ADP bound, while D1 protomers B−F are ATP bound with no evidence for divalent cation as expected (Supplementary Table 2). All D2 protomers are ATP bound and lack divalent cation.

While the spire in the complete sx20S complex reconstruction is ordered enough to distinguish the various components, continuous disorder limits the local resolution of the spire reconstruction. To overcome this, we performed signal subtraction to remove the D1 and D2 rings, followed by local alignment of the α-SNAP−syntaxin subcomplex structure alone (Supplementary Fig. 4, "Methods"). This approach improved the resolution of the reconstruction of the α-SNAP −syntaxin sub-complex to 4.25 Å, revealing a parallel, four-helix

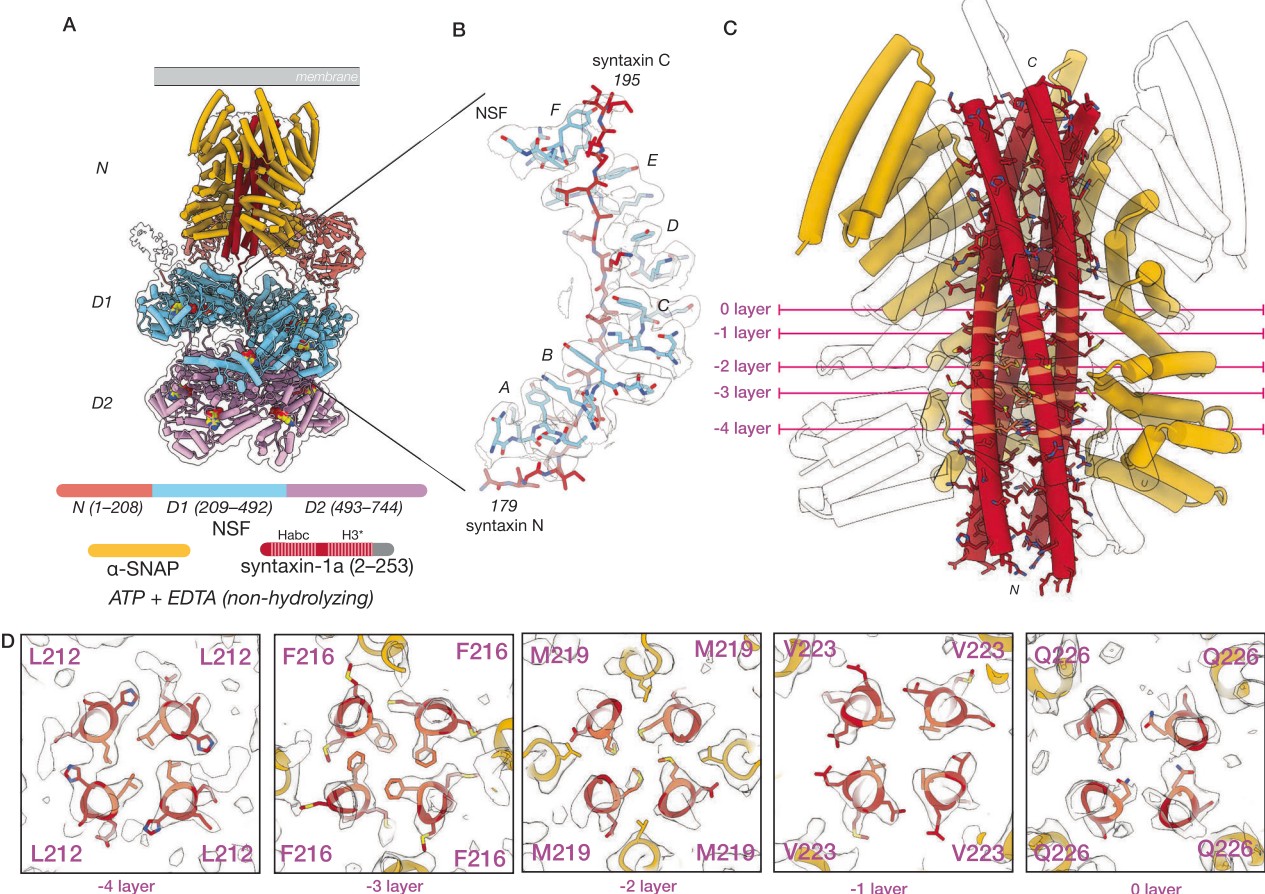

**Fig. 2 | NSF and α-SNAP engage a tetrameric SNARE complex composed of four syntaxin H3 domains.** The structures of the sx20S (class #1, 3.75 Å) and the α-SNAP−syntaxin tetramer subcomplex (class #2, 4.07 Å). **A** Hexameric NSF engages the α-SNAP−syntaxin tetramer subcomplex via its N-domains (salmon), several of which engage four α-SNAP molecules (gold) surrounding the soluble, tetrameric syntaxin H3 complex (dark red) within the catalytic D1 ring of NSF (light blue). The D2 ring of NSF (purple) does not engage the SNARE substrate. A hypothetical membrane is illustrated in gray. In the protein domain legend, ordered syntaxin domains are lightly shaded and those visible in the reconstruction of the SNARE complex are denoted with an asterisk; the syntaxin Habc domains were not observed. **B** NSF (light blue) engages one of the four syntaxin molecules (dark red) in the tetramer via a series of pore loop interactions (protomers A–F, indicated); a tyrosine side chain (Y294) on each of the six protomers intercalates in between every other side chain of a portion of syntaxin N-terminal to its H3 domain. The corresponding sharpened map is contoured at 4 σ. **C** The α-SNAP−syntaxin tetramer subcomplex model, with two of the four α-SNAP models hidden for clarity. The syntaxin H3 SNARE complex forms a twisted, parallel α-helical bundle with a series of key SNARE complex layers preserved. **D** Slices through the α-SNAP−syntaxin tetramer subcomplex model and sharpened map at the -4−0 layers, with key residues lightened for emphasis. Each layer is characterized by nearly planar interactions between four identical side chains within the helical bundle. The primary interactions between α-SNAP and the syntaxin H3 complex are observed near the -2 and -3 layers, where several functionally essential α-SNAP side chains (L197, Y200) are positioned by a loop that buries them within the groove of the bundle. No symmetry was imposed during reconstruction.

bundle of syntaxin SNARE motifs with density of sufficient quality to assign characteristic side chains (Fig. 2C, D). All four helices are aligned in the same register, with a unique hydrophobic layer (−3 layer) composed of four F216 side chains present; remarkably, all fit within the core of the bundle (Fig. 2D, Fig. 5A) and induce little overall deformation of the individual H3 domains (Fig. 5B–D, bottom row). The parallel syntaxin tetramer bundle observed in our structure would allow membrane incorporation of the C-terminal transmembrane domains of all four syntaxin molecules (Fig. 2A).

To determine whether NSF and α-SNAP act to disassemble the syntaxin tetramer, we devised an in vitro, gel-based disassembly assay (Supplementary Fig. 2B, C). Wild-type NSF, α-SNAP, and soluble syntaxin (residues 2–253) were purified independently, mixed in the absence of $Mg^{2+}$ (i.e., non-hydrolyzing conditions), and the sx20S complex was purified. The subsequent addition of $Mg^{2+}$ to the purified complex resulted in a significant loss of α-SNAP−syntaxin following filtration relative to the non-hydrolyzing control, consistent with the disassembly of the syntaxin tetramer and concurrent

disassociation of α-SNAP (Supplementary Fig. 2C, 21± 1% loss). This activity is also shown in the context of the Munc18 capture assay described below (Fig. 1E).

## NSF-catalyzed disassembly of the tetrameric syntaxin SNARE complex is required for efficient syntaxin capture by Munc18

NSF and α-SNAP constantly disassemble SNARE complexes at the synapse, and individual syntaxin molecules can then be captured by Munc18 for efficient priming, and, ultimately, for fusion to occur[3,46,47]. The capture of free syntaxin molecules, either from clusters or otherwise, accompanies the formation of "on-granule" clusters of the syntaxin−Munc18 complex[20,22,25], with implications for animal behavior[48]. Our co-localization results (Fig. 1A–D) suggest that syntaxin is at least partially sequestered in stable SNARE complexes that are a substrate for NSF. This notion is further supported by the finding that α-SNAP must be present for NSF colocalization to occur, as inferred from co-localization experiments with and without the NSF N-domain, the deletion of which precludes α-SNAP binding (Fig. 1A, B). Finally, we

note that α-SNAP itself inhibits fusion by competing with Munc18 to bind syntaxin[30].

Having shown that NSF and α-SNAP disassemble the tetrameric syntaxin SNARE complex (Supplementary Fig. 2B–C), we next performed a related experiment to test the hypothesis that disassembly precedes Munc18 capture by monitoring the formation of the syntaxin−Munc18 complex. Again, wild-type NSF, α-SNAP, and soluble syntaxin (residues 2–253) were purified independently, mixed in the absence of $Mg^{2+}$ (i.e., non-hydrolyzing conditions), and the sx20S complex was purified. The pool was split in half, and Munc18 was then added to each in excess, followed by the addition of either EDTA (non-hydrolyzing control) or $Mg^{2+}$ (hydrolyzing conditions) to initiate disassembly. Following incubation at room temperature for 10 min, size-exclusion chromatography (Fig. 1E) and SDS-PAGE (Supplementary Fig. 2D, E) were performed to assess syntaxin oligomer disassembly and formation of the syntaxin−Munc18 complex[30].

Under non-hydrolyzing conditions, a supercomplex of sx20S−Munc18 formed, with some syntaxin captured by Munc18. Strikingly, the addition of $Mg^{2+}$ triggered nearly complete disassembly of the tetrameric syntaxin SNARE complex, leaving free NSF, 2-fold enriched syntaxin−Munc18, and nearly two-fold enriched free α-SNAP as determined by integrated area (Fig. 1E). Together, these results support the notion that NSF and α-SNAP disassemble the tetrameric syntaxin SNARE complex and show that complete capture by Munc18 is dependent upon this disassembly process, consistent with previous observations of the inhibitory nature of α-SNAP. The finding of a sx20S−Munc18 complex is not surprising, as the N-termini of the syntaxin molecules remain accessible, enabling a weak syntaxin−Munc18 interaction[49].

## Structure of a 2:1 syntaxin−SNAP-25 complex with NSF−α-SNAP

In addition to the syntaxin homo-tetramer, hetero-oligomers with SNAP-25 are expected to be present near or even within syntaxin nanodomains. Indeed, SNAP-25 forms distinct clusters near syntaxin clusters in neurons[19,22,24] and readily forms binary complexes with syntaxin in vitro[31–36,50–52]. Although SNAP-25 is required for fusion, assembly of a fusogenic ternary trans-SNARE complex likely proceeds first through a syntaxin−synaptobrevin intermediate templated by Munc18-1, also found near syntaxin clusters[20,24,25], and by Munc13-1[31,46,53]. Disassembly of the binary complex is critical, as genetic deletion of NSF is lethal[54]. However, an NSF mutation that differentially blocks ternary, but not binary, SNARE complex disassembly causes only intermediate sensorineural defects[38]. Thus, NSF and α-SNAP likely disassemble binary complexes of syntaxin and SNAP-25 in vivo to set the stage for the action of Munc18 and Munc13, ultimately resulting in synchronous and efficient $Ca^{2+}$-triggered fusion[3,31,32,53].

To gain insight into the structural nature of binary complex formation and disassembly by NSF, we prepared a complex of NSF, α-SNAP, and the binary t-SNARE complex (21bin20S) composed of full-length soluble SNAP-25 and soluble syntaxin-1a (1–267) in the absence of divalent cation (i.e., non-hydrolyzing conditions) and performed single-particle cryo-EM (Supplementary Tables 1, 2, Supplementary Fig. 3B, Supplementary Figs. 5, 6). First, models of the NSF D1 and D2 domains with substrate bound in the D1 pore were built into two high-resolution (2.88 Å, 2.92 Å) classes from a single dataset, differing only in the state of F protomer engagement with substrate (Supplementary Fig. 5). Next, the dataset from which they were derived was combined with two additional datasets to enable the 3D reconstruction of weakly populated states with ordered density for the α-SNAP−SNARE spire (Supplementary Fig. 6). Classes with evidence of NSF D1 domains engaging ordered α-SNAP−SNARE sub-complexes were carried forward; eight 21bin20S models were built into reconstructions ranging from 3.39–4.01 Å in resolution, differing mainly in the orientation of N domains and a sub-complex of three α-SNAP molecules and four SNARE motifs relative to the D1 ATPase ring (Fig. 6A). All classes

revealed substrate-bound NSF without $Mg^{2+}$ needed for ATP hydrolysis and disassembly (Supplementary Table 2), and share the same overall architecture, with a spire composed of NSF N-domains and the α-SNAP−SNARE subcomplex on top, the NSF D1 ring in the middle tier, and the NSF D2 ring forming the bottom tier (Fig. 3A). In all cases, the NSF D1 ATPase ring engages SNARE substrate, with one syntaxin molecule bound to the D1 ring pore in most structures (Fig. 3B, Supplementary Table 2).

Like the sx20S complex, continuous variability of the α-SNAP−SNARE subcomplex was present in the 21bin20S reconstructions. We again performed signal subtraction and local alignment on α-SNAP−SNARE sub-complexes from all five classes with ordered spire structures (Supplementary Fig. 6; see "Methods"). This procedure produced a single α-SNAP−SNARE sub-complex reconstruction with an overall resolution of 3.74 Å (Fig. 3C), sufficient to assign characteristic side chains, identify the SNAREs forming the complex and their directionality (Fig. 3D, Fig. 5A). A model composed of the ternary neuronal SNARE complex[55] with synaptobrevin removed was docked into the density, preserving a distinctive kink in the peptide linker leading into the SN2 SNARE motif of SNAP-25. In place of synaptobrevin, a second syntaxin molecule was placed to complete the four-helix bundle with little deviation from its usual backbone configuration of the ternary complex (Fig. 5B). As such, the stoichiometry of the syntaxin−SNAP-25 complex observed is 2:1, the dominant species in solution[33,36,50–52]. Sidechains comprising essential layers repack within the core of the complex (Fig. 3D) compared to the ternary SNARE complex. For example, the ionic central layer (0 layer) is reorganized; Arg56 of synaptobrevin is replaced with a second Gln226 from the additional syntaxin molecule, which adopts an extended rotamer to mimic the arginine it replaces. The SNAP-25 linker, which joins the two SNARE motifs, is largely disordered but blocks binding by a fourth α-SNAP molecule, as observed for the neuronal SNARE complex[6].

Docking the 2:1 binary complex structure back into the seven individual 21bin20S reconstructions reveals an apparent preference for syntaxin, with six of eight reconstructions featuring the N-terminus of syntaxin H3 bound within the pore of the D1 ring (Fig. 3B). As in the case of the 20S with ternary SNARE complex, this observation is consistent with previous work, in which disassembly was not contingent on specific SNARE N-terminus identity[6,7,40]. Taken together, we observe that the NSF D1 ring can bind to at least two different SNAREs, syntaxin, and SNAP-25, in various configurations.

## Structure of a 2:2 syntaxin−SNAP-25 complex with NSF−α-SNAP

To gain insight into the disassembly of the binary SNARE complex, we repeated the binary 20S preparation but added $Mg^{2+}$ and an ATP regeneration system[56] (i.e., hydrolyzing conditions) to the purified complex -1 min before vitrification for single particle cryo-EM (Supplementary Tables 1, 2, Supplementary Fig. 3C, Supplementary Fig. 7). The addition of the regeneration system more closely mimics the metabolic state of the cell, and allows multiple cycles of SNARE disassembly and reassembly to proceed as observed under far more dilute conditions[32]. Following 3D classification, 41% of particles formed a complete complex, all of which have ordered α-SNAP−SNARE spire density and differ largely by spire pose (Fig. 6A). Interestingly, classes with poorly defined SNARE substrate, as observed under the non-hydrolyzing condition, were no longer observed. The remaining 59% of particles were SNARE-substrate free and grouped into three classes with split D1 and D2 rings; presumably, this is a post-disassembly, substrate-released state, and it is not observed under non-hydrolyzing conditions.

Substrate-bound reconstructions reveal a flattened D1 ring in closer proximity to the D2 ring and a strikingly different α-SNAP−SNARE spire structure than observed in the 21bin20S complex under non-hydrolyzing conditions (Fig. 4A, Supplementary Fig. 8A). Again, syntaxin is bound within the pore of the D1 ring (Fig. 4B,

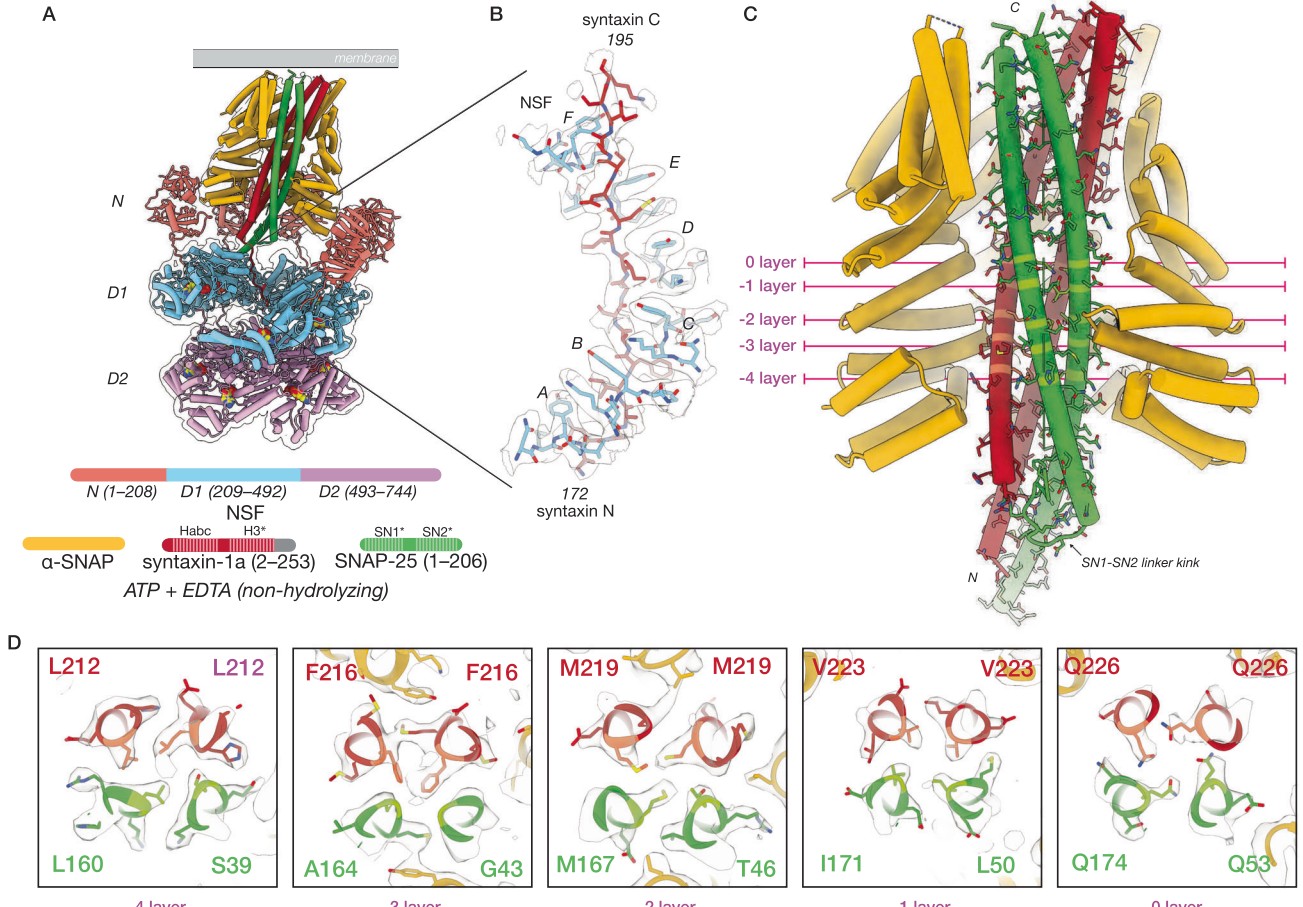

**Fig. 3 | NSF and α-SNAP engage a 2:1 binary SNARE complex of syntaxin H3 and SNAP-25 SN1 and SN2 SNARE domains under non-hydrolyzing conditions.** A representative 21bin20S model (class #5, 3.39 Å) and α-SNAP−2:1 binary SNARE subcomplex model (class #13, 3.74 Å). **A** Hexameric NSF engages the α-SNAP−2:1 binary SNARE subcomplex via multiple N-domains (salmon), which engage three α-SNAP molecules (gold) surrounding the 2:1 binary SNARE complex composed of two syntaxin H3 domains (dark red) and the SN1 and SN2 SNARE domains of a single SNAP-25 molecule (green). A linker N-terminal to one of the syntaxin H3 domains is found within the catalytic D1 ring of NSF (light blue). The D2 ring of NSF (purple) does not engage substrate. A hypothetical membrane is illustrated in gray. In the protein domain legend, ordered syntaxin and SNAP-25 domains are shaded and those visible in the reconstruction of the SNARE complex are denoted with an asterisk; syntaxin Habc domains were not observed. **B** NSF (light blue) engages one of the two syntaxin molecules in the 2:1 binary SNARE complex (dark red) via pore loop interactions involving Y294 (protomers A–F, indicated). A sharpened map is shown, contoured at 5 σ. Of the eight 21bin20S classes with the 2:1 binary SNARE complex from the dataset collected under non-hydrolyzing conditions, six engage syntaxin and two engage SNAP-25 within the pore of the D1 ring. **C** The α-SNAP−2:1 binary SNARE subcomplex model. Three α-SNAP molecules bind the 2:1 binary SNARE complex; a flexible linker between SNAP-25 SN1 and SN2 blocks access by a fourth. The 2:1 binary SNARE complex forms a twisted, parallel α-helical bundle with a series of characteristic SNARE complex layers preserved. This configuration also places the N-terminal portion of the linker, bearing cysteines often palmitoylated in vivo, near the membrane[97]. **D** Layer slices through the α-SNAP−2:1 binary SNARE subcomplex model and sharpened map. Each layer is characterized by nearly planar interactions between four side chains within the helical bundle. The primary interactions between α-SNAP and the syntaxin H3 complex are found near the -2 and -3 layers, where several functionally essential[61] α-SNAP side chains (L197, Y200) are positioned by a loop that buries them within the bundle groove.

Supplementary Table 2). We observed a nearly symmetric α-SNAP− SNARE subcomplex comprising four α-SNAP molecules coating a four-helix bundle. Even without further processing, this structure was recognizable as a parallel 2:2 syntaxin−SNAP-25 sub-complex formed by two syntaxin H3 SNARE domains and two SNAP-25 SN1 SNARE domains; this binary 2:2 configuration was previously observed in crystal structures of syntaxin-1 H3 and SNAP-25 SN1 domains[34] and was unexpected as we prepared the complex with full-length proteins, implying that two independent SNAP-25 molecules contribute their SN1 SNARE domains to this complex. Presumably, disassembly of the binary 2:1 complex (Fig. 3C) had occurred and was followed by reassembly of the 2:2 binary complex (Fig. 4C) before vitrification.

To improve the resolution of the spire reconstruction, signal subtraction and local alignment were performed as before (Supplementary Fig. 7). This approach improved the reconstruction of the spire proteins to 4.22 Å, enabling docking of a crystal structure of the 2:2 syntaxin−SNAP-25 SN1 complex[34] into the density (Fig. 4C), which was of sufficient quality to assign characteristic side chains, to identify the SNARE domains, and to determine their directionality (Fig. 4D, Fig. 5A). The 2:2 binary complex was fit back into the full reconstructions, enabling modeling of the full 22bin20S. Furthermore, we were able to reproduce this result by reconstituting the complex with only the SNARE domains of syntaxin and SNAP-25 under non-hydrolyzing conditions (min22bin20S, Supplementary Fig. 9A–D), yielding the same spire configuration but to a higher resolution of 3.86 Å following local reconstruction (Supplementary Fig. 9E, Fig. 5A). This reconstruction also reveals the interface between an N-domain of NSF and an α-SNAP molecule in detail; the interface is characterized by a core hydrophobic interaction surrounded by charge complementation (Supplementary Fig. 9F).

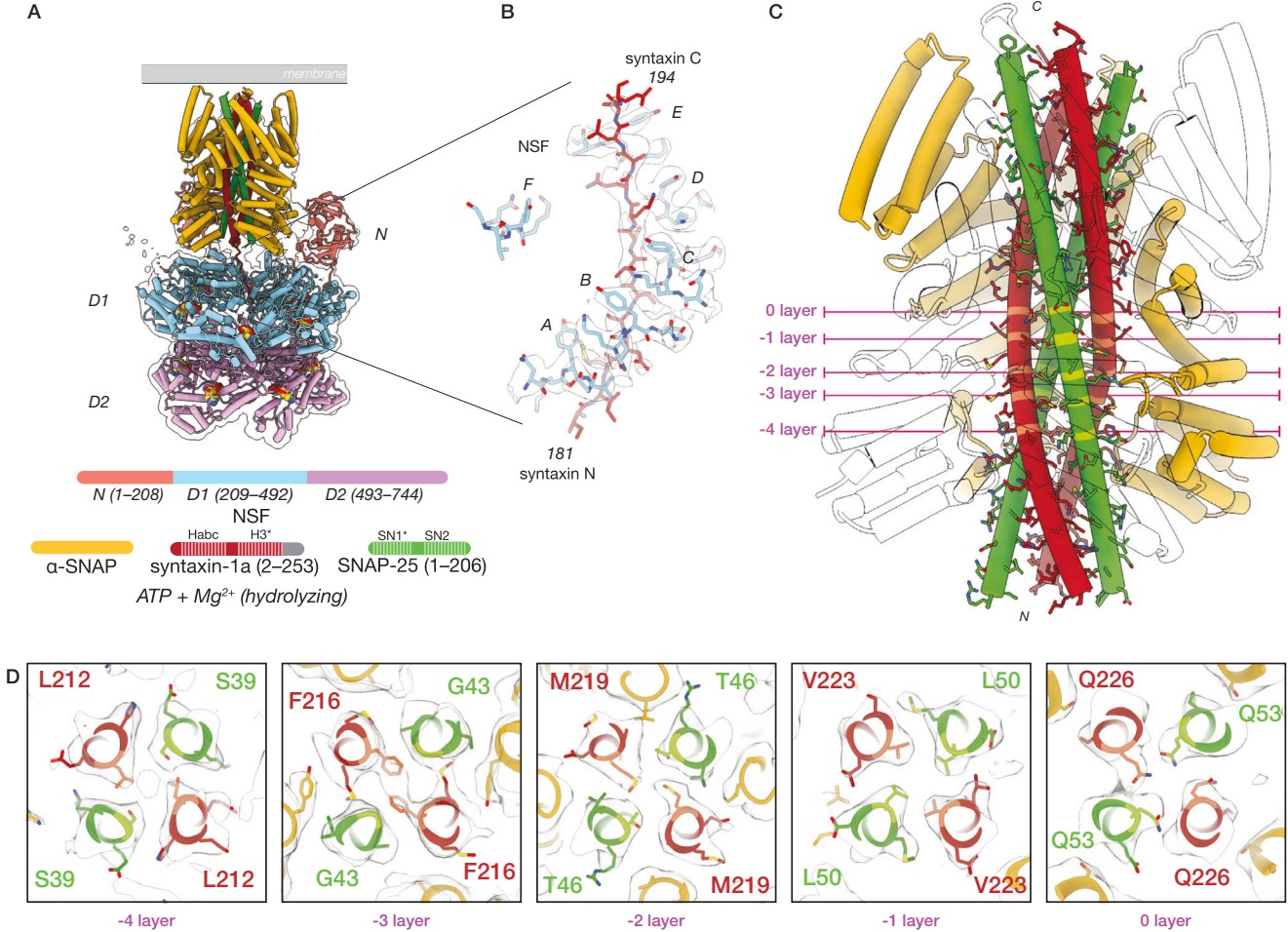

**Fig. 4 | NSF and α-SNAP engage a 2:2 binary SNARE complex of syntaxin H3 and SNAP-25 SN1 SNARE domains under hydrolyzing conditions.** A representative 22bin20S model (class #19, 3.56 Å) and α-SNAP−2:2 binary SNARE subcomplex model (class #23, 4.14 Å). **A** Hexameric NSF engages the α-SNAP−2:2 binary SNARE subcomplex via N-domains (salmon), which engage three α-SNAP molecules (gold) surrounding the soluble, 2:2 binary SNARE complex composed of two syntaxin H3 domains (dark red) and two SNAP-25 SN1 domains contributed by two separate full-length SNAP-25 molecules (green). A portion of a linker N-terminal to one syntaxin H3 domain is observed within the catalytic D1 ring of NSF (light blue). The D2 ring of NSF (purple) shows no substrate density. A hypothetical membrane is illustrated in gray. In the protein domain legend, ordered syntaxin and SNAP-25 domains are lightly shaded and those visible in the reconstruction of the SNARE complex are denoted with an asterisk; syntaxin Habc domains and SNAP-25 SN2 domains were not observed. **B** NSF (light blue) engages one of the two syntaxin molecules (dark red) in the 2:2 binary SNARE complex via pore loop interactions involving Y294 (protomers A–F, indicated). A sharpened map is shown, contoured at 6 σ. Of the five 22bin20S classes with the 2:2 binary SNARE complex, all are found with syntaxin in the D1 pore, while two are observed with the N-terminus of the SNAP-25 SN1 domain in the pore. **C** The α-SNAP−2:2 binary SNARE subcomplex model. Four α-SNAP molecules bind the 2:2 binary SNARE complex, with two SNAP-25 SN1 SNARE domains and two syntaxin H3 domains. This configuration locates two independent N-terminal portions of the SNAP-25 linker bearing cysteine sidechains often palmitoylated in vivo near the membrane[97]. **D** Slices through the α-SNAP−syntaxin tetramer subcomplex at the -4–0 layers. Each layer is characterized by nearly planar interactions between four side chains within the helical bundle. The primary interactions between α-SNAP and the syntaxin H3 complex are found near the -2 and -3 layers, where several functionally essential[61] hydrophobic α-SNAP side chains, L197 and Y200, are buried within the bundle groove. Symmetry was not enforced during local reconstruction. Note that the near-twofold symmetry of the bundle leaves some sidechain densities poorly defined.

## Insights into SNARE substrate engagement and processing by NSF

Hydrolysis by NSF is required for efficient priming and, ultimately, fusion. Collectively, we obtained many datasets with NSF, α-SNAP, and several SNARE substrates under non-hydrolyzing and hydrolyzing conditions (Supplementary Tables 1, 2), presenting an opportunity to investigate the molecular mechanism of SNARE substrate processing and chemomechanical energy transduction in NSF. Together, the resulting high-resolution structures derived from the different classes from these datasets capture the hydrolysis cycle of NSF and reveal important details about its reaction coordinate.

For datasets with particles collected under non-hydrolyzing conditions, NSF is found in two general conformations, primarily differentiated by their F protomer state (Fig. 7A, left column). Two high-resolution 21bin20S classes from the 2:1 binary SNARE complex dataset reveal large-scale conformational displacement connected to the D1 F protomer nucleotide state and SNARE substrate engagement. The first class is ATP-bound and fully SNARE substrate-engaged, while the second has a nucleotide-free and SNARE substrate-disengaged F protomer (Fig. 5A, left, bottom, and top, respectively; Supplementary Fig. 5). The engagement of F at the top of the ring accompanies a relaxation of A into the split. Otherwise, the classes are nearly identical; protomers B–E D1 domains also bind ATP, while protomer A, at the bottom of the D1 split, is ADP-bound in both cases. Comparison of the two classes reveals limited differences outside of the protomers A and F positions, as the change in protomer F position allows additional relaxation of protomer A into the split.

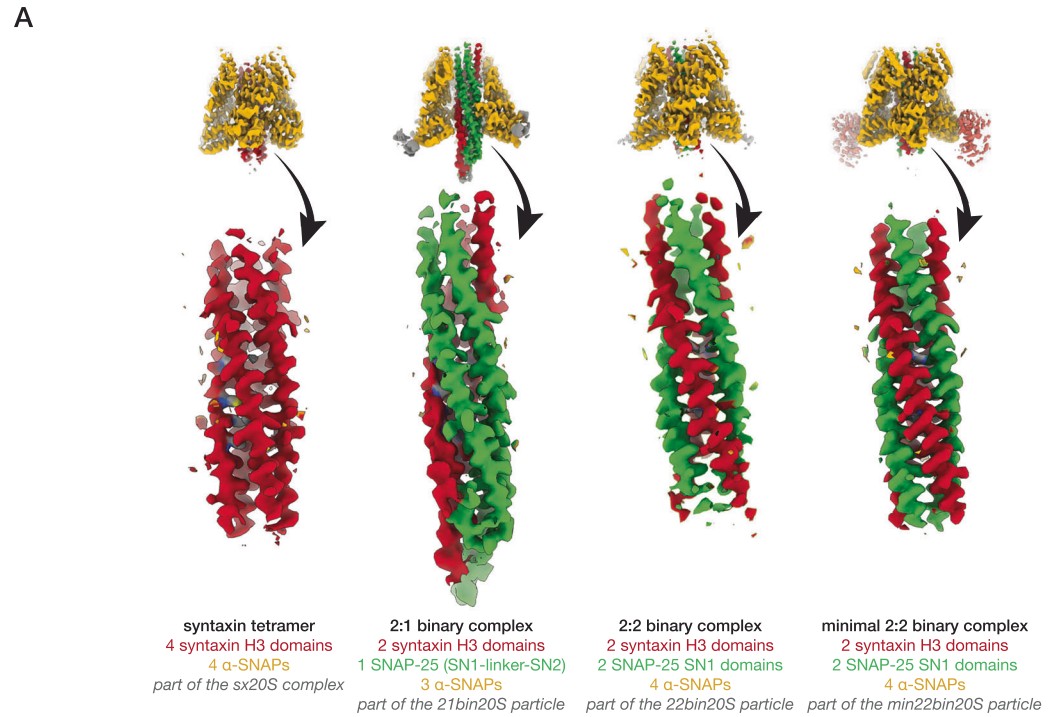

**syntaxin tetramer**
4 syntaxin H3 domains
4 α-SNAPs
*part of the sx20S complex*

**2:1 binary complex**
2 syntaxin H3 domains
1 SNAP-25 (SN1-linker-SN2)
3 α-SNAPs
*part of the 21bin20S particle*

**2:2 binary complex**
2 syntaxin H3 domains
2 SNAP-25 SN1 domains
4 α-SNAPs
*part of the 22bin20S particle*

**minimal 2:2 binary complex**
2 syntaxin H3 domains
2 SNAP-25 SN1 domains
4 α-SNAPs
*part of the min22bin20S particle*

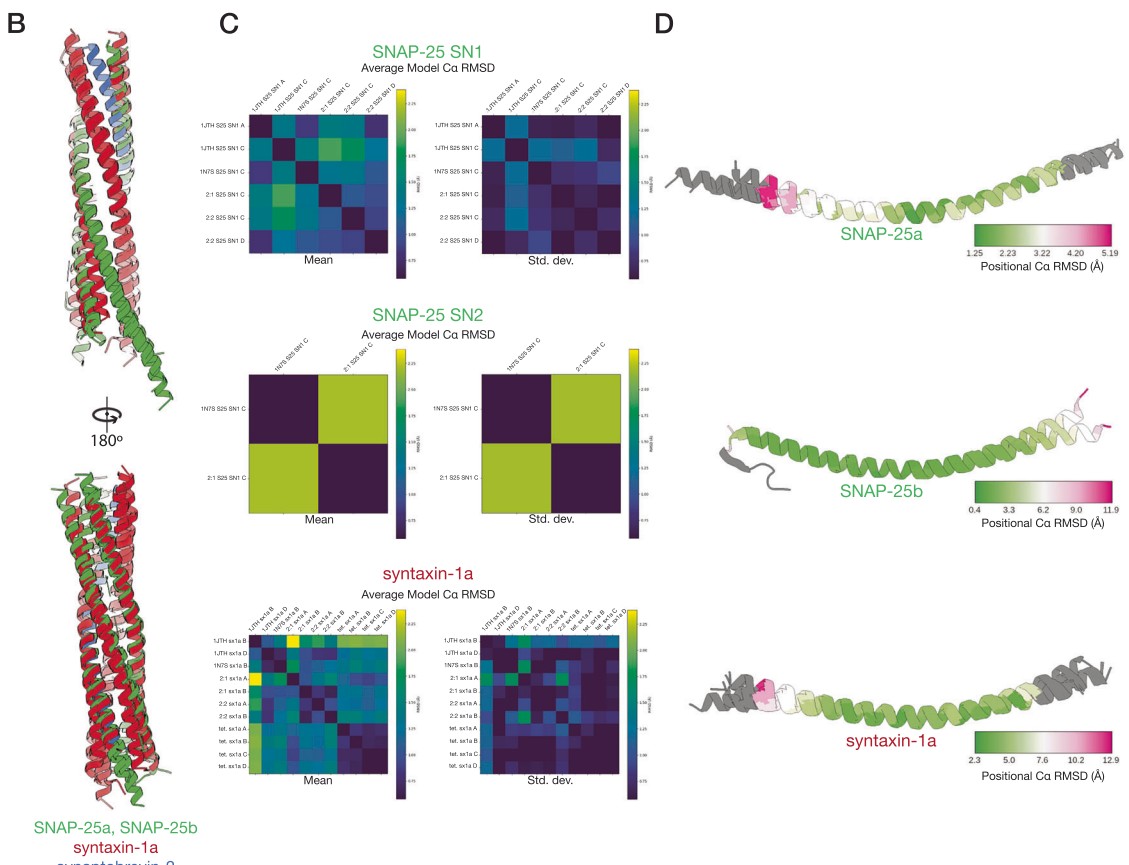

SNAP-25a, SNAP-25b
syntaxin-1a
synaptobrevin-2

In contrast to the non-hydrolyzing datasets, particles in the hydrolyzing dataset fall into two categories—substrate-bound 22bin20S (Fig. 7A, middle column) or substrate-free NSF (post-hydrolysis; Fig. 7A, right column), of which there are ten and three classes, respectively (Supplementary Table 2, Supplementary Fig. 7). The particles show a population shift towards a flatter D1 ring that is closer to D2 relative to the non-hydrolyzing class. This is particularly clear in the case of the SNARE-substrate free classes, which also show an expanded pore (Supplementary Fig. 8A, B). Most strikingly, no SNARE substrate-bound classes show an F protomer as fully engaged as those in non-hydrolyzing classes; in all but one class, F is ADP bound, SNARE substrate disengaged, and sits between engaged A and E

**Fig. 5 | Reconstruction and comparison of SNARE complexes and sub-complexes. A** Signal subtraction and local refinement in *CryoSPARC*[80] produced reconstructions of four different α-SNAP−SNARE subcomplexes. In each case, the density of the SNARE proteins in the core of the subcomplex was sufficiently well-defined to assign the SNARE domain identity, directionality, and register. **B** Crystal structures of the neuronal ternary SNARE complex (PDB ID 1N7S[54]) and the 2:2 syntaxin−SNAP-25 complex (PDB ID 1JTH[34]) were superimposed using UCSF ChimeraX[84] with cryo-EM structures from this work of the tetrameric syntaxin H3 sub-complex from the sx20S, the 2:1 syntaxin−SNAP-25 sub-complex from the 21bin20S complex, and the 2:2 syntaxin−SNAP-25 sub-complex within the 22bin20S complex. These structures show similar quaternary structure, maintaining the same register and similar twist in the central SNARE layers. The SNARE complexes and sub-complexes within their respective 20S complexes are somewhat untwisted at the N- and C-terminal ends relative to the crystal structures of the SNARE complexes alone, likely due to a combination of the absence of crystal packing as well as interactions with α-SNAPs and NSF. **C** Quantitative comparison of individual SNARE domain ternary structure. Each pixel in a pair of matrices represents the mean Cα RMSD (Å, left) and corresponding standard deviation (right) between two instances of a given SNARE domain in two different structures. Only Cα atoms shared by all structures were considered. The same color scale is preserved across all comparisons. The mean Cα RMSD generally varies between 1 and 2 Å depending on the specific pair. **D** Individual Cα RMSDs mapped to their corresponding SNARE domain structures, aligned. Core regions of the complexes are structurally similar, but unwinding near the SNARE domain termini leads to larger deviations, particularly near the N-termini where NSF engages one of the N-terminal segments leading into a SNARE domain (SNAP-25 or syntaxin).

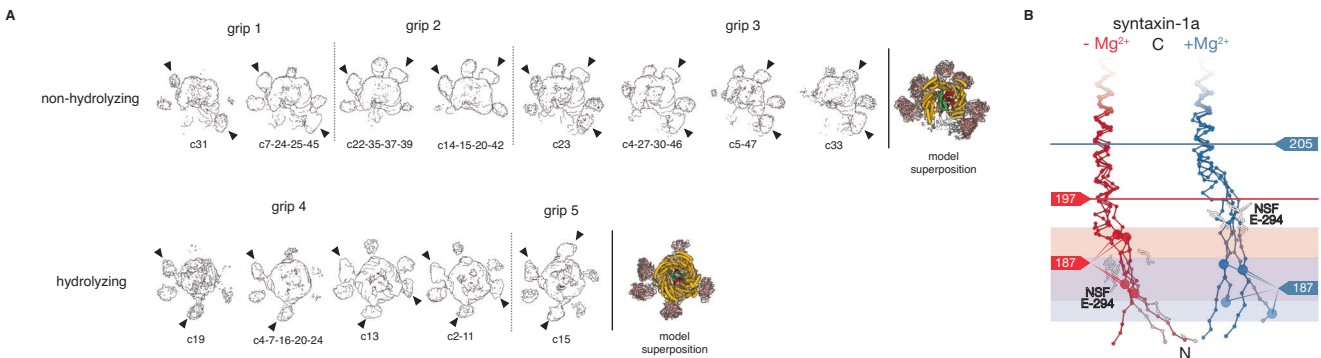

**Fig. 6 | Multimodal engagement of SNARE substrate by NSF. A** The α-SNAP−SNARE subcomplex is engaged by the NSF N domains. The orientation of the subcomplex relative to the NSF D1 ring split, the tilt of the subcomplex relative to the D1 ring, and the specific NSF N-domains engaged to the subcomplex are major determinants of particle class identity in different datasets. Three distinctive "grips" are observed in the case of the classes with the 2:1 syntaxin−SNAP-25 complex identified under non-hydrolyzing conditions (top), and two are observed in the case of the classes with the 2:2 syntaxin−SNAP-25 SNARE complex identified under hydrolyzing conditions (bottom). Pairs of N-domains typically bind a given α-SNAP near its C-terminus; arrows indicate the most well-ordered N-domains as assessed by the presence of two-lobed density. The two cases are not directly comparable, as the 2:1 syntaxin−SNAP-25 sub-complex is bound by three α-SNAP molecules, while the 2:2 syntaxin−SNAP-25 sub-complex is bound by four α-SNAP molecules. This difference may be due to the flexible linker that runs along the 2:1 syntaxin−SNAP-25 sub-complex surface between the SNAP-25 SN1 and SN2 SNARE domains. **B** The SNARE N-termini are engaged by the NSF D1 pore. Variable syntaxin engagement by NSF between non-hydrolyzing and hydrolyzing classes is observed. Alignment of structures with ordered syntaxin substrates in the pore of the D1 ring under non-hydrolyzing (red) and hydrolyzing (blue) conditions reveals disruption of the syntaxin H3 domain up to around residues 197 and 205, respectively.

protomers. However, a subclass with an ATP-bound F protomer was found by subdivision of a large class with 3D Variability Analysis[57] (3DVA; Supplementary Fig. 10C, D) for which the interprotomer interface with the neighboring D1 protomer E is nearly−but not completely−formed. Critically, the pore loop approaches the substrate-engaged state found in the non-hydrolyzing classes; together, this suggests a trajectory of protomer A dissociation, nucleotide release and rebinding, and, finally, SNARE substrate rebinding (Fig. 7D). Furthermore, in classes with ordered syntaxin substrate, the unstructured region of syntaxin N-terminal to the H3 domain is engaged by the D1 pore loops around eight residues further into the primary sequence under hydrolyzing conditions (Fig. 6B).

### Sequential hydrolysis by NSF drives conformational cycling

As noted above, the nucleotide states of the D1 protomers help explain the pattern of engagement with the SNARE substrate (Fig. 7A–C, Supplementary Table 2). How does nucleotide hydrolysis drive this process? Unlike the non-hydrolyzing 21bin20S reconstructions, hydrolyzing 22bin20S reconstructions are only ATP+Mg²⁺-bound and substrate engaged in protomer E, consistent with a sequential hydrolysis mechanism (Fig. 7A, middle column). Except for the rebinding F protomer described above, all other D1 protomers are ADP bound with variability in the presence of $Mg^{2+}$ and inorganic phosphate ($P_i$); in most cases, protomers C and D have ADP•$P_i$ and $Mg^{2+}$, protomer B is usually ADP-bound, and protomer A is always ADP-bound. Finally, substrate-free classes are always ADP bound, even in protomer E; they are flat, have some N-domains packed between the D1 and D2 rings, and are characterized by a split in both the D1 and D2 rings between protomers A and F (Fig. 7A, right column); this is consistent with a SNARE substrate side-loading mechanism proposed for the NSF ortholog in yeast, Sec18[45]. In both hydrolyzing and non-hydrolyzing datasets, all D2 domains are ATP-bound, with $Mg^{2+}$ present in the hydrolyzing condition.

Together, these classes represent key steps in the hydrolysis cycle, revealing the binding of $Mg^{2+}$, hydrolysis of ATP to ADP•$P_i$ at protomer E, and subsequent $P_i$ release down-ring; these states are accompanied by a conformational change in the Walker A, Walker B, and Sensor 1 active site residues in *cis* and the arginine fingers of the down-ring protomer in *trans* (Fig. 7B), and are connected in turn to larger scale conformational change within the D1 domain (Fig. 7C). Critically, $P_i$ release following hydrolysis leads to arginine finger disengagement (Fig. 7B, right two panels) and subsequent relaxation and solvation of the interprotomer interface, enabling the protomer A transition to protomer F and subsequent substrate rebinding.

What triggers hydrolysis at E, and what conformational changes follow? While hydrolysis could proceed spontaneously, this would be inefficient in the absence of an ATP-bound protomer F up-ring to capture accompanying substrate translocation. We hypothesized that some allosteric mechanism likely regulates this process. To identify such a mechanism in an unbiased manner, we performed principal components analysis (PCA) of Cα coordinates of 132 D1 protomers across all datasets followed by hierarchical density-based clustering

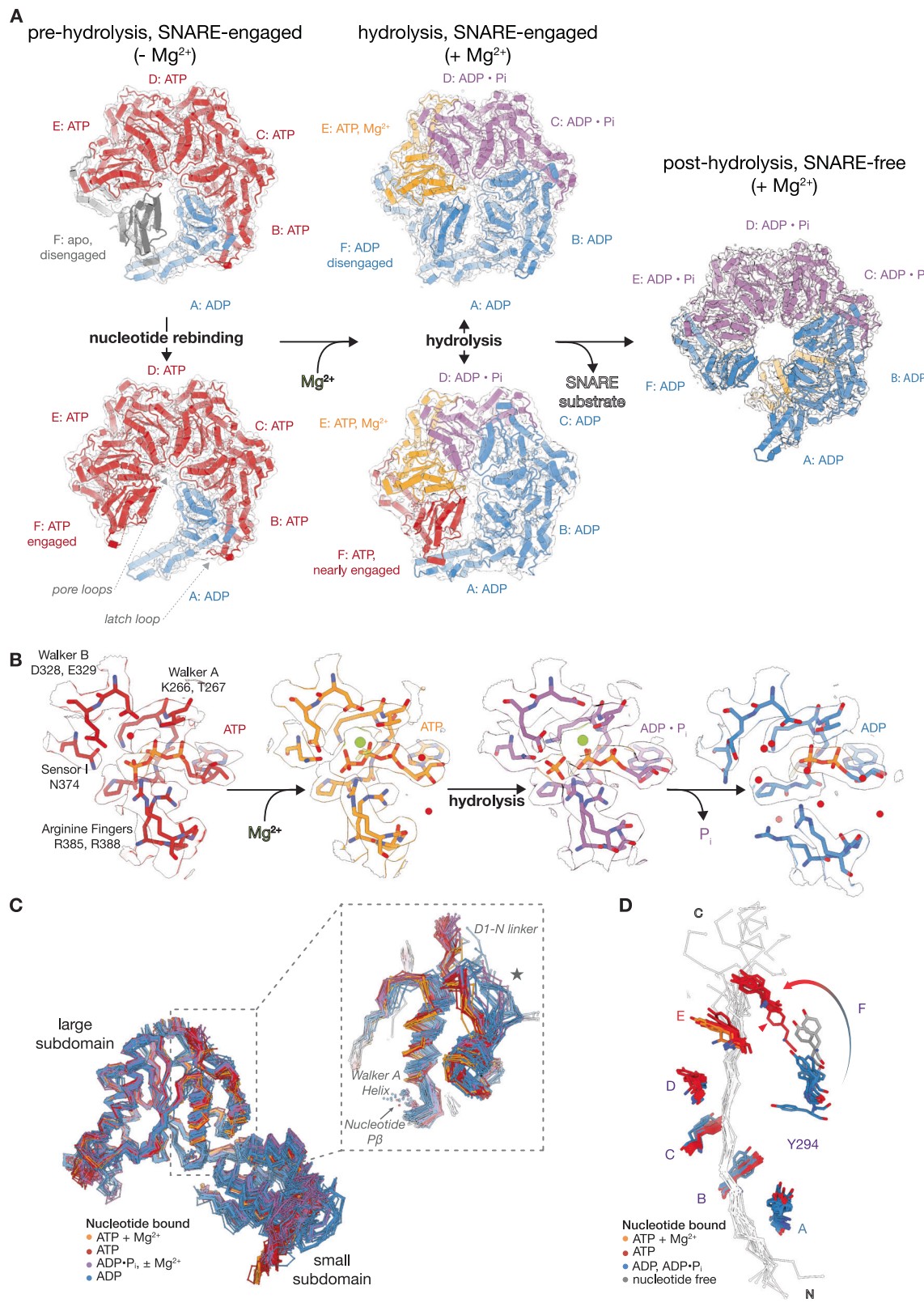

(HDBSCAN) in high-dimensional space, revealing the conformational cycling of the NSF D1 protomers in a low-dimensional coordinate as well as a corresponding allosteric mechanism for modulating hydrolysis by protomers E (Figs. 8, 9).

Overall, we find protomers divided into five interpretable clusters connected to ATP hydrolysis, $P_i$ release, and protomer cycling at the D1

split during SNARE disengagement, nucleotide rebinding, and SNARE re-engagement (Fig. 8A). Here, we focus on the unique features of the protomers E under hydrolyzing conditions in two dimensions of the 11-dimensional latent space, where they cluster far from other protomers (Fig. 8A, transition 1). As noted above, protomers E are the only ATP-$Mg^{2+}$ bound D1 protomers under hydrolyzing conditions (Fig. 8B), and

**Fig. 7 | Comparison of NSF structures under non-hydrolyzing and hydrolyzing conditions reveals the interplay between sequential AAA+ hydrolysis and substrate rebinding. A** Key conformational states of the NSF D1 ATPase ring. Two states dominate in the absence of Mg$^{2+}$ (left); the top class (#4) has SNARE substrate disengaged, nucleotide-free protomer F, while the bottom class (#3) has SNARE substrate-engaged, ATP-bound F. With the addition of Mg$^{2+}$ (middle), E is Mg$^{2+}$-bound. Protomers D, C, and occasionally B retain P$_i$ and Mg$^{2+}$. Protomers A are ADP-bound. The top subclass (#18) has substrate-disengaged, ADP-bound F. The bottom subclass (#17) has nearly substrate-engaged, ATP-bound F. Substrate-free, post-hydrolysis classes, e.g., #24 (right) show an ATP-free D1 ring. Spire, SNARE substrates, and the D2 ring are mostly hidden here. **B** Details of hydrolysis. The P-loop binds nucleotide. Arginine fingers R385 and R388 from the down-ring protomer coordinate nucleotide phosphates. Without Mg$^{2+}$, a water molecule sits between the γ-phosphate and Walker B residue side chains of D328 and E329. Sensor 1 residue N374 is oriented towards the γ-phosphate of ATP. Under hydrolyzing conditions, Mg$^{2+}$ is coordinated by the β- and γ-phosphate moieties of ATP and the

side chain of Walker A motif T267. Following hydrolysis, ADP, P$_i$, and Mg$^{2+}$ remain; Walker B E329 and Sensor 1 N374 side chains coordinate the P$_i$ as it disassociates. P$_i$ release down-ring leads to arginine finger disassociation, disrupting the inter-protomer interface. Models are derived from D1 protomers #4-E, #16-E, #16-D, and #16-B. **C** Alignment of 130 D1 protomer large subdomains from 23 models derived from all datasets. Only Cα and Pβ atoms are shown. The inset shows a key region of the protein that shows coupled changes in the N-D1 linker and the Walker A helix. The latch loop contributed by the up-ring protomer is also coupled to the conformational state of this region. **D** Alignment of tyrosine residues in the pore of the D1 ring of substrate-bound models reveals protomer F as the site of nucleotide exchange and substrate re-engagement. ADP-bound F is disengaged from the substrate (blue). The release of ADP (nucleotide-free, gray) and ATP-binding (red) accompany substrate (re)engagement. The lowest tyrosine corresponding to an ATP-bound F shown (red triangle) is from the model shown in (**A**) (middle column, bottom).

they are uniquely characterized by an extreme large-small subdomain angle (Fig. 8C, Supplementary Fig. 8). Examination of PCA and clustering results reveal conformational states unique to these protomers (Supplementary Fig. 10A, Fig. 8), notably a discrete Walker A (α$_3$) helix position ("down" state) relative to the rest of the D1 large subdomain that uniquely positions the nucleotide near two key catalytic residues (Sensor I, N374 and Arginine finger 2, R388; Fig. 8E left, Fig. 9), a unique conformation of a highly conserved[58] segment of the N-D1 linker connected to N-domain engagement ("up A" state, Fig. 8E, middle), and the absence of a fully engaged "latch loop" (residues 457–467) contributed by the up-ring protomer F in *trans* ("intermediate A" state; Fig. 8E, right). All three of these features appear structurally coupled (Supplementary Fig. 10B) and comprise an allosteric integration mechanism by which the position of nucleotide in the active site is modulated at a distance by both N-domain engagement to SNARE substrate via the N-D1 linker conformation and by the presence of an up-ring protomer F via latch loop engagement. This modulation in turn influences the distance between key catalytic active site residues and nucleotide (Fig. 9).

To test the importance of the latch loop interaction in *trans*, a mutant form of NSF was generated in which a portion of the latch loop (residues 460–466) is replaced with a short glycine-serine linker. An NSF activity assay based on fluorescence dequenching (see "Methods") shows a highly significant, ~72% reduction in the rate of SNARE complex disassembly ($p = 0.0005$; Fig. 8D). On the other hand, the functional importance of the N-D1 linker has already been established. A mutation to NSF in the most conserved region N-D1 linker, I209N, has a direct behavioral effect in zebrafish; NSF(I209N) is defective in the disassembly of ternary SNARE complexes but not binary SNARE complexes, explaining this phenotype[38,58]. This isoleucine side chain directly contacts the engaged latch loop and is adjacent to S207, a putative phosphorylation site[59], further raising the possibility of post-translational modification in regulating NSF activity[60]. Together, these observations support an allosteric integration mechanism critical in modulating NSF function.

## Discussion

Using structural and biophysical studies, we have discovered a key organizational principle at the active zone in the presynapse that likely involves the remodeling of syntaxin clusters. Synaptic vesicle fusion and neurotransmitter release are among the most central processes of synaptic transmission, and understanding the organization and the molecular mechanisms of these processes in the cellular context is essential. Our work helps to answer a key unanswered question—what is the purpose of phase-separated syntaxin at the presynapse?

Our observation of NSF colocalization to syntaxin clusters in vivo, coupled with direct evidence of disassembly of both syntaxin tetramers and syntaxin–SNAP-25 complexes, explains a previously

overlooked but essential aspect of the synaptic vesicle fusion cycle with broad implications for membrane trafficking (Fig. 10A). While NSF is well known for its role in ternary *trans*-SNARE complex disassembly following fusion, this work, combined with functional studies[3,31,32,38,53], further establishes NSF as a critical factor for synaptic vesicle priming before fusion. NSF is enriched at and near syntaxin clusters, which are, in turn, found at the presynapse. This enrichment is highly dependent upon the presence of the NSF N-domain (Fig. 1D), which binds α-SNAP, which in turn binds parallel, four-helix SNARE complexes through a stereotyped binding interface[6,7,61]. The observation here of a parallel, four-helix syntaxin-1a bundle suggests that these clusters are structured, at least in part, and that NSF and α-SNAP act to disassemble the syntaxin tetramers within or on the perimeter of a given cluster, increasing syntaxin mobility at the synapse[27]. Individual syntaxin molecules liberated from clusters by NSF can then be captured by the SM protein Munc18 and ultimately be used to assemble fusogenic *trans*-SNARE complexes.

However, a high local concentration of SNAP-25 is present near the syntaxin clusters as well[11,24,62], so free syntaxin may also enter a non-productive binary SNARE complex with SNAP-25 with a stoichiometry dependent upon local concentrations and configurations of either protein. NSF disassembly of these species prior to fusion is critical in vivo[38]. In this work, we observe binary complexes with two stoichiometries—2:1 and 2:2 binary complexes of syntaxin and SNAP-25, with the latter re-forming following NSF-mediated disassembly of the former under hydrolyzing conditions. The 2:1 configuration dominates in the non-hydrolyzing condition—and in other in vitro studies[33,51]—but is derived from co-expressed proteins that were extensively purified without NSF action. The 2:2 configuration is only found following NSF-mediated disassembly of the 2:1 configuration. This striking observation is likely related to differences in the effective concentration of syntaxin dimers preceding complex formation during expression in and purification from *E. coli* versus following disassembly in vitro[52]. Regardless, these structures are consistent with the notion that SNAP-25 SN1 is a critical initial mediator of SNARE assembly[63]. The structural plasticity of the SNARE complexes is also of note; taken together, the repeated observation of a twisted, parallel, four-helix bundle with minimal backbone distortion (Fig. 5) emphasizes the functional conservation latent in the SNARE domain sequences studied thus far, as well as the importance of additional SNARE protein accessory domains in differentiating SNARE protein function[45]. Thus far, only parallel SNARE bundles have been observed the context of various 20S complexes in this work and elsewhere[6,7,32,61]. While NSF and alpha-SNAP may rapidly disassemble anti-parallel SNARE complexes[32], such complexes are apparently too rare or heterogeneous to be observable by cryo-EM.

We note that several recent studies[20,48] also highlight the critical connection between syntaxin clustering, neurotransmission, and

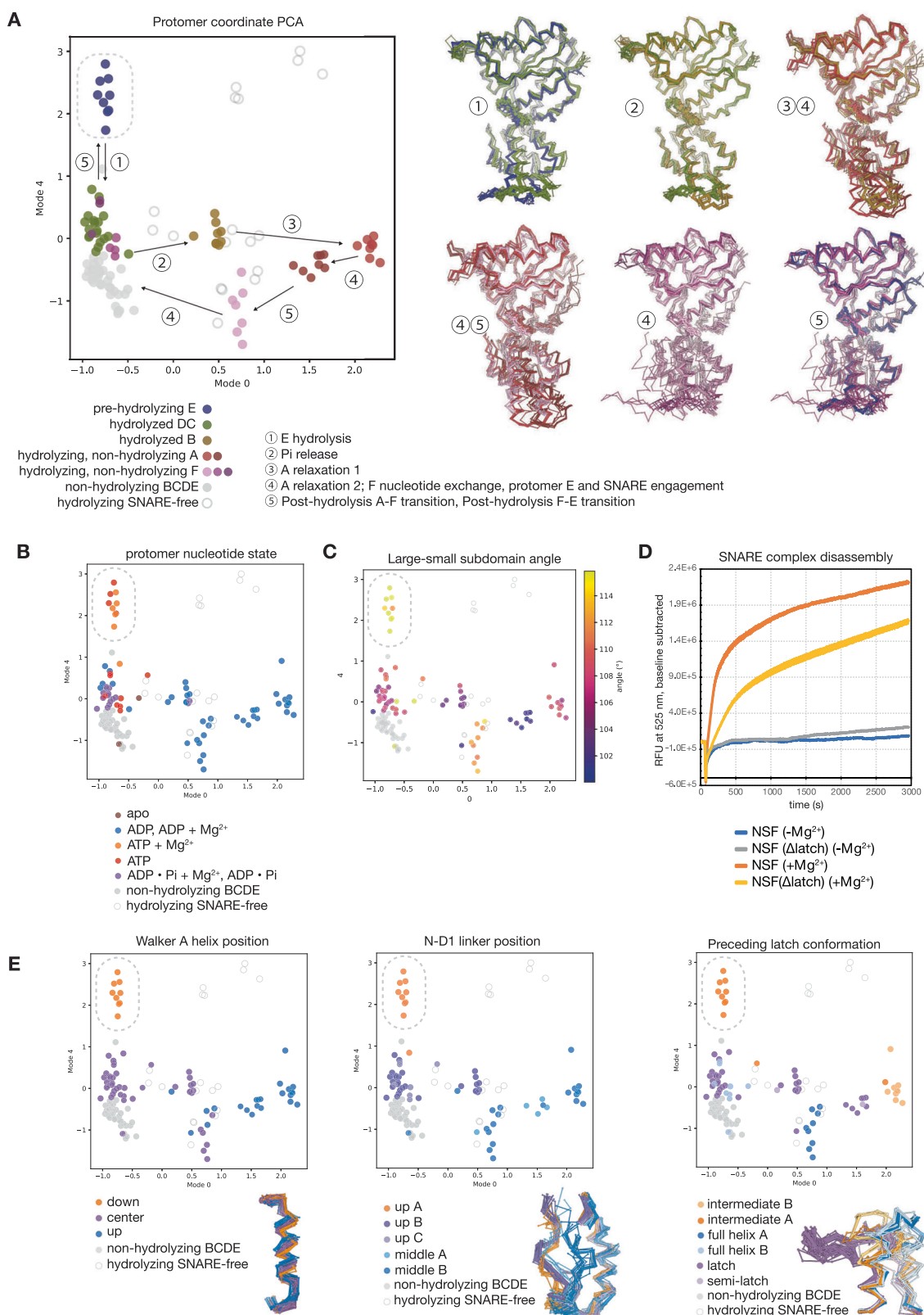

animal behavior and emphasize the role of the protein Munc18 in capturing free syntaxin from clusters for subsequent priming. Our disassembly results show that syntaxin capture by Munc18 depends on its liberation from SNARE complexes (Fig. 1E) on the timescales of synaptic transmission. This is also consistent with previous observations of increasing syntaxin mobility at the synapse with NSF action[27]. As such, a complete mechanistic picture of the system involves NSF, α-SNAP, and Munc18; Munc18 can directly bind any monomeric syntaxin at or near the syntaxin clusters, but it likely cannot actively remove syntaxin from oligomeric complexes within the clusters, especially on the timescales of synaptic transmission. Our model also suggests a mechanism for the transit of syntaxin from off-pathway, Munc18-free clusters to on-granule clusters of syntaxin-1a−Munc18[20,22] and further explains the inhibition of Munc18 binding of syntaxin by α-SNAP[30].

**Fig. 8 | Principal component analysis of NSF D1 protomer coordinates reveals quantitative mechanistic details of sequential hydrolysis. A** 132 D1 domain models were aligned by large subdomain, subjected to Cα coordinate PCA, and clustered with HDBSCAN in 11-dimensional space (left). Protomers cluster into discrete groups based on conformation, visible upon projection into a plane defined by the first and fifth modes. A proposed hydrolysis cycle is indicated. Superimposed models corresponding to these transitions are shown (right). The cluster corresponding to pre-hydrolysis protomers E is circled. **B** Protomer nucleotide state, mapped to the 2D PCA coordinate from (**A**). The ATP + Mg²⁺ state is unique to protomers E (circled). **C** The angle (°) between helices α₃ and α₈ (a proxy for the relative angle between the D1 large and small subdomains). Protomers E have the largest angle (circled). **D** A mutant form of NSF, NSF(Δlatch), in which part of the latch loop (residues 460–466) was replaced with a short glycine-serine linker (SGS), was assayed for its ability to disassemble the ternary SNARE complex composed of syntaxin, SNAP-25, and synaptobrevin[6]. Traces show the mean ± SEM of

eight replicates for each form of NSF, with and without adding MgCl₂. Disassembly rates were calculated by linear regression to each trace in the linear region of the time course, from -95–350 s. NSF(Δlatch) disassembled SNARE complex at 28 ± 1% the rate of wild type. **E** Three conformational features distinguish the pre-hydrolysis protomer E cluster (circled). These features are in contact and structurally coupled. The position of nucleotide in the active site is modulated by the position of the Walker A helix (α₃), which occupies three discrete states; pre-hydrolysis protomers E occupy the "down" state (left). The position of the α₃ is allosterically modulated by at least two inputs. The N-D1 linker conformation (center) is connected to the state of N-domain binding to α-SNAP, and it interacts with the latch loop of the up-ring protomer upon rebinding. The latch loop of the up-ring protomer contacts this region in *trans* and is also structurally coupled (right). These two elements push against the C-terminal end of α₃ and ultimately modulate the position of the ATP γ-phosphate relative to key catalytic residues.

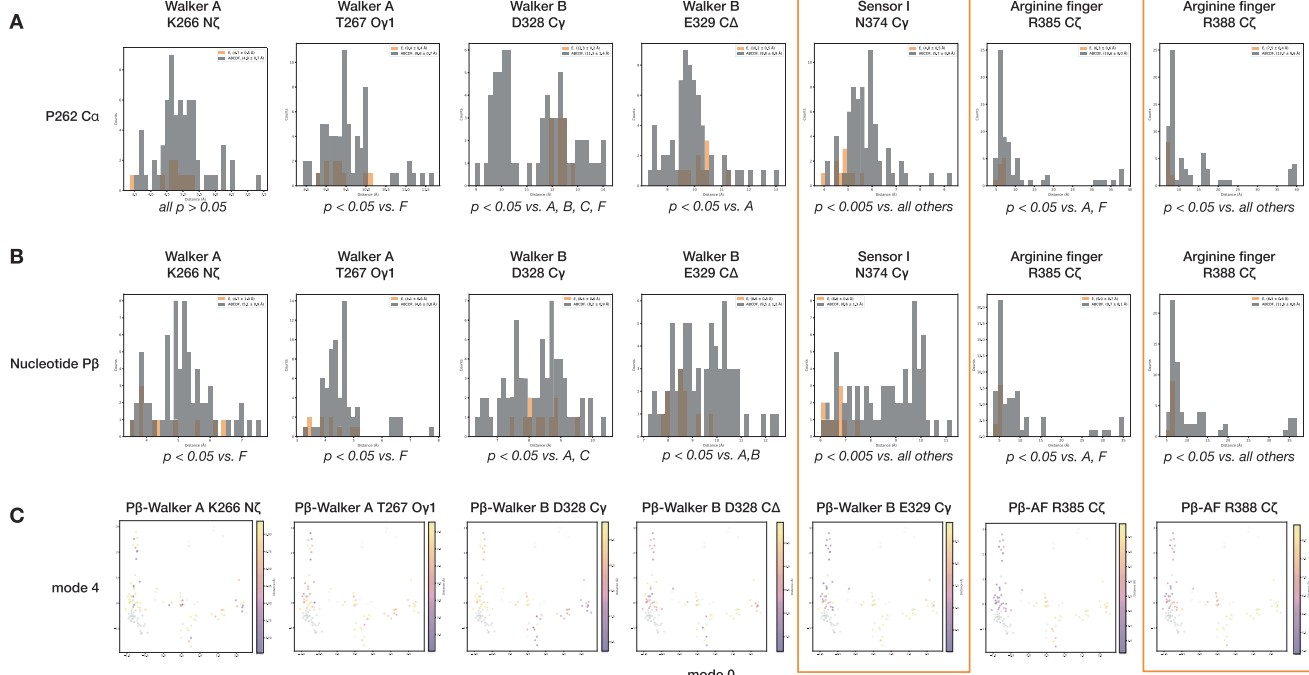

**Fig. 9 | The pre-hydrolysis state of protomers E has several unique active site features. A** Distance distributions between the P262 Cα and a given active site residue. P262 Cα is a relatively invariant reference atom for measuring distances, as it is well-constrained by local geometry and the presence or absence of Pγ, Pᵢ, or Mg²⁺ in the active site. Protomers E (orange bars) have a significantly closer Sensor I N374 distance to P262 (4.8 ± 0.5 Å vs. 5.7 ± 0.9 overall, $p < 0.005$ for all pairs) than any other protomer (black bars). The terminal carbon of the second arginine finger, R388 Cζ, was also significantly closer in protomers E than in any other protomer (7.5 ± 0.4 Å vs. 13.7 ± 9.6 Å, $p < 0.05$ for all pairs). While not a specific observation to protomer E, the P262 Cα-Walker B D328 Cγ distance distribution is bimodal, with ADP-bound protomers generally having a smaller distance between the two atoms. Welch's *t*-test, assuming unequal variances, was used to assess significance in all cases. **B** Distance distributions between the Pβ atom of either ATP or ADP and

a given active site residue. Pβ was used as it is found in all nucleotide-bound protomers. The pattern of significant differences between protomers E and protomers A–D and F described for P262 Cα is mostly repeated here, with the Sensor I N374 average distance to Pβ found to be highly significant (6.6 ± 0.4 Å vs. 8.6 ± 1.3, $p < 0.005$ for all pairs). As in (**A**) the terminal carbon of the second arginine finger, R388 Cζ, was also significantly closer to Pβ in protomers E than in any other protomer (6.3 ± 0.6 Å vs. 11.6 ± 8.8, $p < 0.05$ for all pairs). Some differences are present in other cases. Aside from specific concerns related to protomers E, many of these distributions also show a bimodal pattern consistent with different nucleotide states. Welch's *t*-test, assuming unequal variances, was used to assess significance in all cases. **C** Distances from the histograms in (**B**) mapped to the coordinate determined by PCA.

Regardless, NSF and α-SNAP likely disassemble both syntaxin tetramers and binary SNARE complexes to maintain a fusogenic population of *t*-SNAREs[3,38], which ultimately enter the *trans*-SNARE complex with the *v*-SNARE synaptobrevin-2 and set the stage for Ca²⁺-triggered fusion in cooperation with synaptotagmin and complexin[64,65]. Indeed, the energy released by SNARE protein zippering during vesicle fusion can be directly traced to the hydrolysis of ATP by NSF, which effectively maintains them in a high-energy state. Thus, syntaxin nanodomains could act as high-density storage sites for syntaxin, and NSF could tap these stores under periods of high-

frequency vesicle release. Modulation of NSF activity by post-translational modification could thus be one way to regulate this process[38,60]. Finally, given the observation of clustered Sso1p in yeast, and the high degree of structural and functional conservation between NSF and Sec18[45], this seems likely to be a conserved organizing principle across fusion contexts.

At the atomic level, how does NSF accomplish the energy transfer needed to disassemble various SNARE complexes? The structural details of the hydrolysis cycle (Fig. 7) are consistent with sequential hydrolysis by the six D1 ATPase domains of NSF. Hydrolysis drives

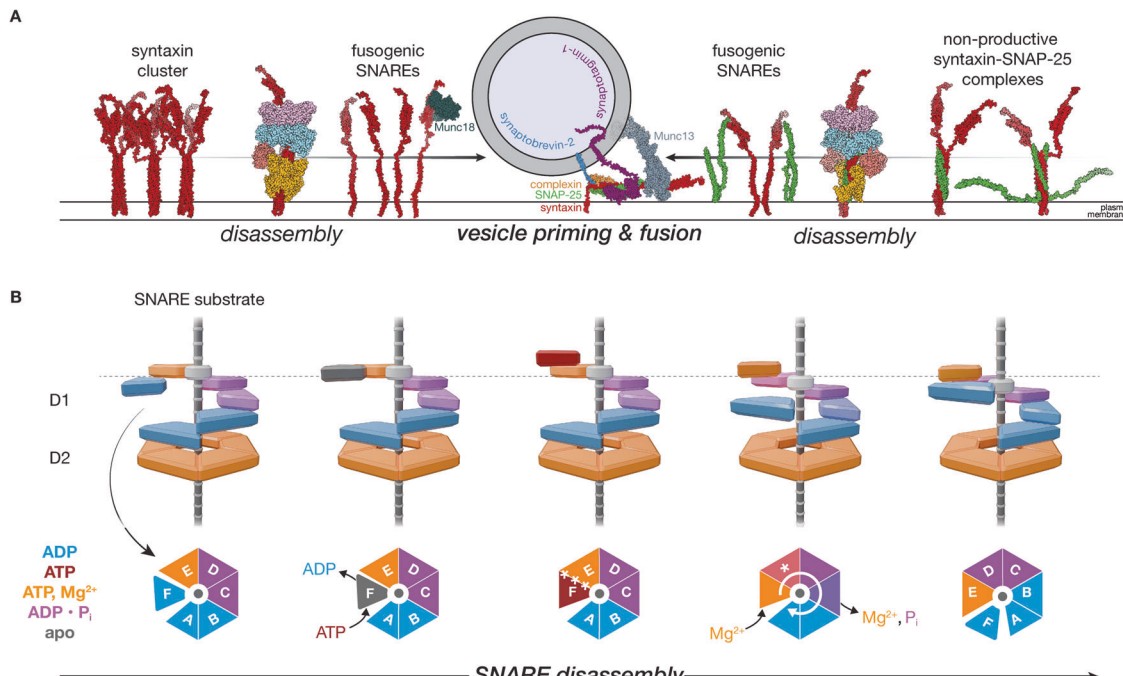

**Fig. 10 | An expanded model for NSF action across cellular and atomic scales.**
**A** A model for disassembling non-fusogenic SNARE species by NSF and α-SNAP at the presynapse and progression to priming and fusion. Syntaxin nanodomains comprised at least in part of tetrameric syntaxin SNARE complexes serve as a reservoir for the liberation of individual, fusogenic syntaxin molecules by NSF upon disassembly. These syntaxin molecules may also enter non-productive binary complexes with SNAP-25, with 2:1 or 2:2 stoichiometry; such complexes would be disassembled by NSF to ensure *trans*-SNARE complex formation with synaptobrevin on incoming synaptic vesicles in concert with other factors, such as Munc18, Munc13, complexin, and synaptotagmin. Where possible, full-length structures (structures reported here and PDB IDs 1N7S, 4JEU, 5W5C) were supplemented with AlphaFold2[94] structures to provide additional context, and ISOLDE[85] was used with secondary structure restraints to reposition flexible regions. **B** A sequential hydrolysis model for SNARE complex disassembly by NSF is inferred from comparing different classes from non-hydrolyzing and hydrolyzing conditions. Following hydrolysis-independent substrate side loading, the D1 ring is found in a pre-hydrolysis state with ATP in protomer E of D1. Protomer F nucleotide exchange then enables ATP exchange and rebinding to both SNARE substrate and the formation of the interprotomer interface with protomer E, triggering ATP hydrolysis in protomer E and substrate translocation. This is accompanied by $P_i$ and $Mg^{2+}$ release from down-ring protomers; in many structures, the release occurs between protomers C and B. The cycle then begins anew. Blue corresponds to ADP-bound, red to ATP-bound, orange to ATP+$Mg^{2+}$ bound, purple to ADP•$P_i$ bound, and gray to nucleotide-free. See Movie S1 for an animated rendition.

---

conformational change, which is coupled to the application of force to at least one member of the SNARE complex, destabilizing it and promoting disassembly over multiple hydrolysis events (Fig. 10B, Supplementary Movie 1). Rather than relying on a non-physiological nucleotide state promoted by chelation of divalent cations or non-hydrolyzable nucleotide analogs (e.g., AMP-PNP), we established a reaction condition more closely approximating cellular metabolism by using a physiological ATP regeneration system. Our sequential hydrolysis model suggests that hydrolysis proceeds around the asymmetric NSF D1 ring, occurring near the "top" of the ring at protomer E. Down-ring protomers reveal evidence for a transient post-hydrolysis reaction state in which ADP•$P_i$ is generally observed before $P_i$ release from protomers C or B and an associated disruption and solvation of the interprotomer interface. This mechanism is likely conserved in other members of the AAA+ family, such as p97 (Supplementary Discussion).

Furthermore, the continuous transition from protomer A at the bottom of the ring to protomer F at the top accompanies ADP release and ATP rebinding (Fig. 10B, Supplementary Fig. 10D, E, Supplementary Movie 1). This cycling appears coupled to the triggering of hydrolysis at protomer E in a process allosterically mediated by several key structural elements (Figs. 8, 9); presumably, full protomer F rebinding triggers hydrolysis, likely driven by an allosteric shift associated with full latch engagement with the N-D1 linker region, further formation of the interprotomer interface, and mediation of the distance between nucleotide and key catalytic residues such as Sensor I and the second Arginine finger

(N374 and R388, respectively; Fig. 9). Protomer F rebinding is also connected to the mechanism by which engagement of the SNARE substrate is maintained during processing. In the absence of a completely protomer F-rebound state in the hydrolyzing condition, however, we cannot distinguish between SNARE substrate rebinding before or after protomer E hydrolysis; the former case (ATP in fully rebound F and E) represents a classical "hand-over-hand" mechanism, while the latter case (ATP in fully rebound F, ATP ADP•Pi in E), a ratchet mechanism in which F re-engages substrate during or immediately following a translocation event driven by protomer E hydrolysis, locking in substrate translocation[66]. The flattened D1 rings of the SNARE-substrate-free classes with ADP•Pi in protomers E (Supplementary Fig. 8) are suggestive of the latter; flattening upon hydrolysis at protomer E could exert a pulling force on the substrate, and translocation would then take place with substrate engagement by a new protomer F and release of the old protomer A. Multiple rounds of translocation are expected to ultimately lead to melting of the engaged SNARE domain and subsequent disruption of the SNARE complex. These large-scale conformational changes may also apply a force via the NSF N-domains, which could aid in the disassembly process[6,7,61].

Together, our findings expand the role of NSF in synaptic transmission and the membrane fusion cycle while further demonstrating its contribution as a model system for understanding energy transduction and macromolecular remodeling by AAA+ proteins. NSF acts as a mobile engine of the presynapse—working together with the SNAP adapter proteins and factors such as Munc18 to convert the chemical

energy latent in ATP into the physical work needed to organize, maintain, and drive synaptic function through the disassembly of a variety of SNARE proteins.

## Methods

### PC12 transfection and imaging

The PC12 cells, generously provided by Prof. Tobias Meyer from Stanford University, were cultured on glass-bottomed dishes (MatTek; P35G-1.5-14-C) and allowed to adhere for 24 h. To introduce genetic material into the cells, transfection was performed using the TurboFect™ Transfection Reagent (#R0532, Thermo Fisher Scientific, USA) according to the manufacturer's instructions. In brief, the plasmids (NSF-GGGSGGGS-mScarlet-I in pTwist CMV Puro, 1.75 μg; syntaxin1a-eGFP plasmid, 0.5 μg) were combined with 6.75 μL of TurboFect in 225 μL of DMEM medium (without FBS) and incubated for 20 min. The DNA-TurboFect mixtures were carefully added drop-wise to the cell cultures and incubated for 6 h. Following the incubation, the cell cultures were exchanged into complete culture medium. After 24 h post-transfection, the cells underwent three washes with PBS. Subsequently, the cell membranes were imaged using a structured illumination Nikon N-SIM system (version AR5.11.00 64 bit, Tokyo, Japan) equipped with an Apochromat 100×/1.49 numerical aperture oil-immersion objective lens. Using a 488 nm laser, the samples were excited to acquire green SIM images, which captured emission within the 500–550 nm bandwidth, specifically for GFP. Additionally, red SIM images were obtained by exciting the samples with a 561 nm laser to capture emission within the 570–640 nm bandwidth, corresponding to mScarlet-I. To determine the degree of colocalization between syntaxin and NSF, Pearson's correlation coefficient was calculated between channels *ImageJ*[67].

### Yeast imaging

The Sso1-GFP strain was created previously[68] and provided by the lab of Daniel Jarosz; the yeast harbors a plasmid with Sso1-GFP similar to that used in the PC12 experiments, in which GFP is found outside of the cell while Sso1p itself is in the plasma membrane and cytoplasm. BY4741 is the parental strain; N-SWAT Sso1 GFP is the protein, and **MATa his3Δ1 leu2Δ0 met15Δ0 ura3Δ0** is the genotype. Expression was carried out using the native yeast promoter NOP1. Super-resolution microscopy of *S. cerevisiae* expressing Sso1-GFP was performed using structured illumination microscopy (SIM) on the OMX-Blaze system (Applied Precision, Issaquah, WA). Glass-bottom dishes (1.5H, Cellvis) were pretreated with 0.01% poly-l-lysine by adding 600–1000 μL to the center and incubated overnight. Following incubation, the dishes were washed with deionized water to remove excess poly-l-lysine. Yeast cultures were grown overnight, washed twice with phosphate-buffered saline (PBS), and resuspended to an optical density (OD) of 0.4. A 10–20 μL volume of the yeast suspension was deposited onto the treated dishes, followed by the addition of Diamond Antifade (Thermo Fisher Scientific) to prevent photobleaching during imaging. The glass-bottom dishes were placed in a metal holder with 1–2 drops of immersion oil (refractive index 1.512) added before placement on the microscope stage. SIM imaging was conducted using a laser appropriate for GFP excitation, and images were captured with a 100× objective for a z-slice thickness of 0.125 μm, spanning a total thickness of 2.6 μm. Widefield images were also collected at 0.2 μm slice thickness when necessary.

### Protein expression and purification

All NSF, the wild-type and mutant NSF-mScarlet-I, and NSF(Δ460–466) proteins were derived from NSF from the Chinese hamster *C. griseus*, expressed in BL21(DE3)-RIL *E. coli*, and purified as described previously[6], with final reassembly of hexameric NSF in reassembly buffer (50 mM Tris pH 8.0, 150 mM NaCl, 1 mM EDTA, 1 mM ATP, and 1 mM TCEP). α-SNAP from rat *R. norvegicus* was

expressed in *E. coli* and purified as described previously[69]. Soluble rat neuronal SNARE complex for the fluorescence-based disassembly assay (wild-type SNAP-25A, syntaxin-1A 1–265, and 6×His-synapto-brevin-2 1–96) was purified by co-expression in C41 *E. coli* and purified as described previously[69] Purified soluble rat neuronal SNARE complex was fluorescently labeled with Oregon Green 488 maleimide dye as described previously[6].

Syntaxin-1A fragment 2–253 (Stx$^{2-253}$) was N-terminal His$_6$-tagged using the plasmid pTEV5. Overexpression in *E. coli* BL21(DE3) was carried out in 8 L of auto-inducible media at 37 °C for 4 hr and then 30 °C overnight. Cells were collected by centrifugation; half of the pellet was resuspended into 200 mL of 50 mM TRIS pH 8, 500 mM NaCl, and 10 mM imidazole and supplemented with DNAse, lysozyme, 1 mM MgCl$_2$, and 2 cOmplete Protease Inhibitor Cocktail tablets (Roche). Cells were lysed by sonication (1 s on/3 s off, total 20 min, 55% amplitude, 4 °C). Debris were removed by centrifugation at 43,000 rpm for 30 min at 4 °C using a Ti45 rotor. The supernatant was then mixed with 5 mL of Ni-NTA agarose resin (Qiagen) equilibrated with the lysis buffer. The suspension was gently stirred for 1 h at 4 °C. The resin was subsequently recovered by centrifugation at 4200 rpm for 10 min at 4 °C, poured into a column, and attached to an AKTA Start (Cytiva). An extensive wash was performed using 20 CV of 50 mM TRIS pH 8, 500 mM NaCl, and 40 mM imidazole until the 280 nm signal was stable. Protein was eluted in 1 mL fractions in 50 mM TRIS pH8, 200 mM NaCl, and 400 mM imidazole. Syntaxin-enriched fractions were identified by SDS-PAGE and pooled. To remove the His-tag, the protein was mixed with an excess of TEV protease and dialyzed against 20 mM HEPES pH 7.5, 50 mM NaCl, 1 mM DTT, overnight at 4 °C. The sample was centrifuged at 4000 rpm for 10 min at 4 °C, filtered, and loaded onto a Mono-Q 5/50 column attached to an AKTA Pure (Cytiva). The resin was extensively washed with 20 mM HEPES pH 7.5, 50 mM NaCl, 0.2 mM EDTA until 280 nm signal was stable. Elution was performed in 120 CV over 2 hr from 0.050–1 M NaCl. 2 mL fractions within the major peaks were run on SDS-PAGE, and fractions containing Stx$^{2-253}$ were concentrated using a 10 kDa cutoff centrifugal filtration device. The concentrate was filtered and further purified using a Superdex Increase 75 16/60 column (Cytiva) equilibrated with 20 mM TRIS pH 8, 150 mM NaCl. Fractions were run on an SDS-PAGE, and fractions containing Stx$^{2-253}$ were concentrated using a 10 kDa MWCO centrifugal filtration device. A final concentration of 745 mM was determined by using the extinction coefficient 5960 M$^{-1}$ cm$^{-1}$. Stx$^{2-253}$ was aliquoted, flash frozen, and stored at −80 °C until further use. An independent sample purified from the same batch of cells was analyzed on MALDI-TOF, and a monomeric molecular weight of 29,208.50 g/mol was confirmed.

Binary neuronal SNARE complex for cryo-EM imaging studies (SNAP-25A with all cysteines mutated to serine, syntaxin-1A 1–267) was purified following co-expression of SNARE proteins from *E. coli* BL21 STAR (Invitrogen). Cells were grown overnight at 30 °C using an auto-inducing LB medium[70]. After harvesting the cells by centrifugation, the pellet was resuspended in lysis buffer (50 mM NaPi pH 8.0, 300 mM NaCl, 20 mM imidazole, 0.5 mM TCEP) supplemented with lysozyme, DNase I, and cOmplete Protease Inhibitor Cocktail tablets (Roche) and subjected to sonication (3 s on, 9 s off, 60% power for 5 min, repeated 3×) and centrifugation (10,000 rpm for 10 min followed by 43,000 rpm for 40 min). The clarified lysate was bound to Ni-NTA agarose beads (QIAGEN) equilibrated in a lysis buffer. Beads were harvested by centrifugation (2200 rpm for 10 min), poured into a gravity column, and washed with lysis buffer, urea buffer (50 mM NaPi pH 8.0, 300 mM NaCl, 30 mM imidazole, 0.5 mM TCEP, 7.5 M urea), and wash buffer (50 mM NaPi pH 8.0, 300 mM NaCl, 30 mM imidazole, 0.5 mM TCEP). The SNARE complex was then eluted with wash buffer supplemented with an additional 320 mM imidazole. The eluent was supplemented with tobacco etch virus (TEV) protease and dialyzed against buffer A1 (50 mM Tris-HCl, pH 8.0, 250 mM NaCl,

0.5 mM TCEP) overnight at 4 °C. After dialysis, the sample was diluted to a final NaCl concentration of 50 mM using buffer A1 without NaCl, and the His-tag-free complex was subjected to anion exchange chromatography (Buffer A2: 50 mM Tris-HCl, pH 8.0, 50 mM NaCl, 0.5 mM TCEP; Buffer B1: 50 mM Tris-HCl, pH 8.0, 500 mM NaCl, 0.5 mM TCEP) using a MonoQ 4.6/100 column (Cytiva) and a linear gradient of NaCl starting at 50 mM and ending at 500 mM. The peaks containing the purified complex were pooled and loaded onto a Superdex 75 16/60 column to normalize NaCl concentration (50 mM Tris-HCl pH 8.0, 50 mM NaCl, 0.5 mM TCEP).

The sx20S supercomplex was formed by mixing ATP-bound NSF, syntaxin, and α-SNAP to a ratio of 1:5:30 in reassembly buffer and purifying it by size-exclusion chromatography using a Superose 6 10/300 GL column; the first peak to elute was collected and concentrated to around 15 mg mL$^{-1}$ for vitrification and cryo-EM imaging. The 21bin20S supercomplex was purified similarly but with a 1:5:30 ratio of NSF, binary SNARE complex, and α-SNAP. In the case of preparing 21bin20S supercomplex for imaging under hydrolyzing conditions, EDTA was omitted from the final SEC buffer.

### Size exclusion chromatography (SEC)-based syntaxin disassembly and Munc18 capture assay

The sx20S complex was prepared as described in the "Protein expression and purification" "Methods" Section without EDTA in the reassembly buffer and with a 1:50:30 ratio of ATP-bound NSF, syntaxin, and α-SNAP. The NSF and α-SNAP concentrations were informed by the fraction of sx20S to NSF in the cryo-EM experiment (Supplementary Fig. 4) and resulted in an increase of the fraction of 20S with the tetrameric syntaxin SNARE complex relative to free NSF. Following concentration to 50 mg mL$^{-1}$ (or around 66 μM assuming full sx20S formation, although the actual concentration is likely lower), the sx20S sample was split into two 35 μL pools and resuspended to 500 μL in 464 μL of 43 μM Munc18, for a final concentration of 5 μM sx20S (likely lower) and 40 μM Munc18. Either 1 μL of 500 mM EDTA or 1 μL of 500 mM MgCl$_2$ was then added to the pool, in the latter case initiating the disassembly reaction. Following incubation for 10 min at room temperature, the pools were independently subjected to size exclusion chromatography using a Superdex 200 Increase 10/300 GL column (Cytiva), and fractions were analyzed by SDS-PAGE. Absorbance data for each run were processed using the script *elution.py*. Data parsed using *pandas*[71] and *numpy*[72], smoothed using Savitzky-Golay filtering as implemented in *scipy*[73] with a window of 1000 points and a polynomial order of 2, and peaks with absorbance amplitudes greater than 10 mAU, distances greater than 1 mL, and prominences greater than 0.5 mL were found using *find_peaks* in *scipy*. Peaks were integrated using the *trapz* function as implemented in *numpy*. Data were plotted using *matplotlib*[74].

### Filtration-based syntaxin disassembly assay

Soluble syntaxin (2–253) and α-SNAP were mixed at concentrations of 100 μM and 150 μM, respectively, in 50 mM Tris pH 8.0, 150 mM NaCl, 1 mM TCEP, and 1 mM EDTA (SEC buffer) and incubated for 20 min. NSF was subsequently added to a final concentration of 2.5 μM, bringing the total reaction volume to 500 μL. The mixture was then injected into a Superdex 200 16/60 column (Cytiva) and complex peak fractions were pooled and concentrated using a 100,000 MWCO filtration device. For each assay condition, 5 μg of complex was used. Two experimental conditions were performed in triplicate; either EDTA or MgCl$_2$ was added to a final concentration of 4 mM. Samples were incubated for 20 min at room temperature and re-concentrated using 100k MWCO filters before SDS-PAGE. Re-concentration effectively de-enriches liberated syntaxin and α-SNAP molecules. Densitometry was performed using *ImageJ*[67], and statistical significance was established between the conditions using a two-tailed Student's *t*-test.

### Gel-based ternary SNARE complex disassembly assay

To verify that NSF activity was not hampered by the C-terminal addition of mScarlet-I, SNARE complex disassembly was evaluated by gel electrophoresis. Note that the ternary neuronal SNARE complex runs as a single band unless boiled in SDS buffer, or disassembled by NSF, in which case the individual SNARE components are enriched as the complex band disappears. NSF, NSF-mScarlet-I, α-SNAP, and the neuronal SNARE complex were dialyzed into 25 mM TRIS pH 8, 150 mM NaCl, 1 mM ATP. Then, components were combined into either EDTA buffer (25 mM TRIS pH 8, 2 mM ATP, 1 mM EDTA) or Mg$^{2+}$ buffer (25 mM TRIS pH 8, 2 mM ATP, 4 mM MgCl$_2$). The reaction volume was 10 μL, and contained 500 nM NSF or NSF-mScarlet-I and 50 nM α-SNAP. Assays were started by adding 10 μM SNARE complex, and samples were then incubated at 37 °C for 10 min in a thermocycler. 2 μL of 5×-Laemli loading buffer was then added, and samples were incubated at 95 °C for 5 min. Finally, 5 μL of each reaction was loaded onto an AnykD TGX gel (BioRad) and run at 200 V for 45 min. The gel was stained with Coomassie R-250 (BioRad). As internal controls, all single components were diluted at a similar concentration as the reactions using Mg$^{2+}$ buffer and treated in parallel. Note that the additional control marked as "SNARE complex unboiled" corresponds to the SNARE complex incubated at 37 °C but not at 95 °C after adding loading buffer. For reference, monomeric NSF is ~83 kDa while NSF-mScarlet-I is ~112 kDa.

### Fluorescence dequenching-based neuronal SNARE complex disassembly assay

The details of the fluorescence dequenching assay have been previously described[6,7,32,38,40,69]. Fluorescence dequenching assays were carried out using a FlexStation II 384-well plate reader (Molecular Devices) with a final reaction volume of 60 μL in 20 mM Tris pH 8, 100 mM NaCl, 2 mM ATP, and 0.5 mM TCEP. The assay plate was first prepared by adding 50 μL of 60 nM wild-type or Δ460–466 mutant NSF, 2.4 μM α-SNAP, 480 nM OG-labeled neuronal SNARE complex in reaction buffer to each well. Then, 10 μL of reaction buffer (as a control) or ATP hydrolysis initiation buffer (reaction buffer supplemented with 24 μM MgCl$_2$) was added to appropriate wells of the compound plate, and fluorescence dequenching was monitored. Excitation was performed at 485 nm, and emission was monitored at 525 nm with a dichroic mirror at 515 nm. First, fluorescence intensity was monitored before reaction initiation for 60 s. Then, 10 μL of compound plate solution was transferred to each assay plate well (diluting all protein 5/6-fold and MgCl$_2$ concentrations six-fold) and triturated, initiating the disassembly reaction. The final protein concentrations following mixing were 50 nM NSF, 2.0 μM α-SNAP, and 400 nM SNARE complex. The final MgCl$_2$ concentration was 4 mM. Dequenching was then monitored for 30 min at 20 °C. Linear regression over the first several minutes was performed to estimate relative disassembly rates. The statistical significance of changes to disassembly rates was assessed using the two-tailed Student's *t*-test.

### Sample vitrification for cryo-EM SPA

The samples were prepared immediately before vitrification (Supplementary Fig. 3). Quantifoil R1.2/1.3 200 mesh gold grids (Quantifoil Micro Tools GmbH, Germany) were cleaned with chloroform, dried for several hours, and subjected to glow discharge using a PELCO easiGlow Glow Discharge Cleaning System (Ted Pella) under vacuum at 15 mA for 45 s with a 15 s hold time. Immediately following 21bin20S or sx20S supercomplex purification, proteins were concentrated to ~15 mg mL$^{-1}$ in 17 μL. For the hydrolyzing condition, 2 μL of 10 × ATP Regeneration Solution (50 mM Tris pH 8.0, 150 mM NaCl, 1 mM ATP, 160 mM creatine phosphate, and 0.3 mg/mL rabbit muscle creatine kinase (Roche)) was added, and the sample was incubated on ice for 2 min; in contrast, 2 μL of reassembly buffer was added in the case of non-hydrolyzing conditions. Then, for the hydrolyzing condition, 1 μL of 20× Reaction Initiation Solution (50 mM Tris pH 8.0, 150 mM NaCl,

1 mM ATP, 20 mM MgCl₂, 1% Nonidet P-40) was added, otherwise 1 μL of 1% Nonidet P-40 was added in the case of non-hydrolyzing conditions. Using a Vitrobot Mark IV (Thermo Fisher Scientific), 4 μL of a given sample was applied to a grid at room temperature and 100% humidity blotted for 4 s with a blot force of 3, and subsequently plunge-frozen in liquid ethane.

## Cryo-EM SPA data collection

Grids were imaged using a Titan Krios electron microscope (Thermo Fisher Scientific) operated at 300 kV and equipped with a K3 camera (Gatan). Data collection was performed using *SerialEM*[75]. All datasets were collected with a nominal magnification of 22,500× for a physical pixel size of 1.096 Å and a super-resolution pixel size of 0.548 Å. The dose rate was either ~10 or ~20 e⁻/px/s depending on whether correlated double sampling (CDS) was enabled on the K3, for a total dose of around 30 e⁻/Å². Nine grid holes were imaged for each stage movement using the multiple record setup in *SerialEM*, with one collection per hole. At each hole, either a 2 s or 4 s exposure fractionated into either 40 or 80 frames was collected, depending on whether CDS was enabled. Defocus values ranged from −1 to −3 μm. See Supplementary Table 1 for details.

## Cryo-EM SPA image processing

All cryo-EM datasets were processed using a software pipeline consisting of *MotionCorr2*[76], *CTFFIND4*[77], *RELION 3*[78,79], and *cryoSPARC*[57,80]. Early-stage processing was carried out in *RELION 3*, with final 3D refinement and variability analysis carried out with *CryoSPARC*. Dataset-specific processing schemes are presented in Supplementary Figs. 4–7 and Supplementary Fig. 9. Generally, all processing proceeded as follows. Processing began in *RELION 3*, with per-micrograph optics groups defined for a given dataset based on the nine beam shift positions using the custom script *update_relion_tilt_class_og.py*. For example, the hydrolyzing condition micrographs were collected in two sessions, so 18 total optics groups were defined. The raw image stacks were dose-weighted and motion-corrected using the *RELION* implementation of *MotionCorr2*, with seven patches along the x direction and five patches along y. CTF parameters were estimated for each micrograph using *CTFFIND4* with a resolution range of 3–30 Å, an FFT box size of 1024 px, and otherwise default parameters. Micrographs were then selected based on maximum resolution and the CTF figure of merit.

Generally, an initial round of manual picking, extraction, and 2D classification were used to assess data set quality and provide particles to reconstruct an initial 3D model. Good particles from manual picking were then used for automated particle picking in *RELION*. Particles were then extracted with a box size of 640 px, binned 8× (4.384 Å Å pixel size) to a box size of 80 px, and subjected to at least one round of 2D classification to eliminate obviously bad particles. Next, multiple rounds of 3D classification with a 280 Å mask diameter and a regularization parameter $T = 4$ were performed over many iterations to determine the proper number of discrete states in a given dataset, and to ensure that particles were classified properly. Convergence of classification was monitored with a custom script, *startle.py*. Several classes slightly larger than the number of discrete states were used, and classes that appeared to lie along a continuous conformational coordinate were subsequently re-merged into a single class. In the case of the hydrolyzing dataset, SNARE substrate-bound and SNARE substrate-free particles were split into separate groups of particles, re-classified, and processed independently.

Following classification, particles were re-centered and re-extracted with a box size of 588 px and binned 2× (1.096 Å pixel size) with a box size of 294 px. Following an initial round of 3D refinement and post-processing, CTF refinement was performed. Typically, defocus, astigmatism, B-factor, tilt, and trefoil were refined on a per-particle or per-micrograph basis, depending on the specific class. Particles were then subjected to an additional round of 3D refinement and post-processing, and optimal parameters for particle polishing were determined by training on several classes, yielding different parameter sets. Particle polishing was then performed with these sets, and 3D refinement and post-processing were used to assess the results. Polished particles were then subjected to a final round of 3D refinement, post-processing, and CTF refinement, during which anisotropic magnification was estimated. At this stage, particles and 3D reconstructions were imported into *CryoSPARC* for a final round of non-uniform refinement, which typically improved the quality of reconstructions.

In the case of one class observed under hydrolyzing conditions (c4-7-16-20-24), a large degree of continuous heterogeneity was observed. 3D variability analysis (3DVA) revealed a trajectory of protomer F nucleotide rebinding and substrate engagement, so particles in the first mode were split using a custom script, *split_by_3dva.py*, as current tools to do so natively were unavailable at the time. This script fits a normal distribution to the eigenvector magnitude distribution and allows slicing with different cutoffs. For the 3DVA subclasses (slices O and P), particles in the ±1.5 σ tails of the distribution were used for reconstruction.

These reconstructions were considered final for the overall 20S architecture, but further processing was required to generate local reconstructions of the various α-SNAP−SNARE subcomplexes. In the case of the sx20S dataset, only one class was observed with ordered spire density, so local refinement was relatively simple. *CryoSPARC* signal subtraction (*particle_subtract*; masks generated using *Segger* 2.5.3[81] in *Chimera* 1.15[82]) was used to remove the signal from NSF N, D1, and D2 domains, leaving only the α-SNAP−syntaxin H3 SNARE domain tetramer subcomplex. Local refinement (*new_local_refine*) was then performed on these particles with a 40°/20 Å search extent and a custom fulcrum at the N-terminal end of the SNARE bundle near the point of interaction with the F protomer of the D1 ring. The resulting volumes were re-centered on the center of mass (*volume_alignment_tools*), and particle alignments were updated as well (*align_3D*). Then, a final round of local refinement (*new_local_refine*) was performed with the map center as the fulcrum (the default setting), a smaller search extent of 10°/5 Å, and a maximum alignment resolution of 0.1°. Per-particle scales were also reset, and minimization was performed over per-particle scale. This produced the final reconstruction of this subcomplex for subsequent modeling. In the case of the α-SNAP−2:1 and α-SNAP−2:2 binary SNARE subcomplexes, this procedure was followed for each class meeting certain criteria (number of particles, ordered spire, resolution), yielding five local reconstructions for the α-SNAP−2:1 binary SNARE subcomplex and six local reconstructions for the α-SNAP−2:2 binary SNARE subcomplex. In each case, these reconstructions were re-centered (*volume_alignment_tools*), aligned to one another with the center of mass at the origin (*align_3D*), and a final round of local refinement was run to produce a final higher-resolution reconstruction incorporating data from multiple classes. This produced the final reconstruction of these subcomplexes for subsequent modeling. For more details about all data sets, see Supplementary Figs. 4–7, Supplementary Fig. 9, Supplementary Tables 1, 2, and Supplementary Fig. 11.

## Model building and refinement

All manual model building and refinements were carried out using *Coot*[83]. Semi-manual refinement into low-resolution features was performed with *UCSF ChimeraX*[84] and the *ISOLDE*[85] plugin. Automated refinement was performed using *phenix.real_space_refine*[86,87]. No composite maps were used at any stage of model building or refinement.

First, models of the three α-SNAP−SNARE subcomplexes were built by rigid body fitting of the best possible starting models into their corresponding densities using *UCSF ChimeraX*. In each case, 3–4

copies of a high-quality α-SNAP model from the previous best 20S model (PDB ID 6MDM[6]) were rigid body fit into the density. In the case of the syntaxin H3 domain tetramer complex, four copies of the syntaxin chain from the high-resolution crystal structure (PDB ID 1N7S[55]) of the neuronal SNARE complex were docked into the density, with several characteristic residues as fiducials. The quality of the local reconstructions of the SNARE−α-SNAP spires allowed unique assignment of the identity of the SNARE domains, their directionality, and register. In the case of the 2:1 binary SNARE complex, the syntaxin and SNAP-25 components of the same crystal structure were docked into the density, but with the SNARE domain of synaptobrevin replaced with an additional syntaxin H3 SNARE domain; key side chain densities as well as a distinctive kink leading into the SNAP-25 SN2 domain guided placement (Fig. 3C, Fig. 5A). In the case of the 2:2 binary SNARE complex, a previous crystal structure (PDB ID 1JTH[34]) was docked into the density without modification. In all cases, individual components were trimmed, or residues were added where supported by the density. A round of *phenix.real_space_refine* was then performed with only rigid-body refinement enabled for each component. Subsequent cycles of manual model building were performed in *Coot* and *ISOLDE* followed by 1–7 macrocycles of *phenix.real_space_refine* with global minimization and local grid search enabled. Ramachandran and secondary structure restraints were used throughout. Note that while sequence information was included upon deposition for the full-length construct used in the case of each SNARE, some regions (e.g., the syntaxin Habc domain, the SNAP-25 SN2 domain in the case of the 2:2 binary SNARE complex) were not visible in the reconstructions and were not modeled as a result.

Reference models of NSF were then built. Gold-standard models were first built into two high-resolution NSF classes from a single non-hydrolyzing dataset with near-atomic resolutions using the previous best model of NSF (PDB ID 6MDM[6]) as a starting model. The starting model was first docked into cryo-EM density using *ChimeraX*. Next, a round of *phenix.real_space_refine* was performed with only rigid-body refinement enabled; each large and small subdomain of each protomer was treated as a single rigid body (twelve total). Nucleotides and water molecules were then added where appropriate. Subsequent cycles of manual model building were performed in *Coot* and *ISOLDE* followed by 1–7 macrocycles of *phenix.real_space_refine* with global minimization and local grid search enabled. Ramachandran and secondary structure restraints were used throughout.

The previously described reference models were then used as starting models in building the sx20S and bin20S supercomplexes. In sx20S and bin20S reconstructions with well-defined N-domains (i.e., a recognizable two-lobed density), one or more instances of the reprocessed model derived from the crystal structure (PDB ID 1QCS[88]) as re-processed by *PDB_REDO*[89] were rigid body fit into density before refinement. Next, the appropriate α-SNAP−SNARE subcomplex was docked into the density, completing the spire structure. Spires in the global 20S reconstructions were generally heterogeneous; although secondary structure could often be distinguished, side chain density could not, leading to poor map-model correlations in these regions. A round of *phenix.real_space_refine* was performed with only rigid-body refinement enabled; each large and small subdomain of each protomer, each N domain, and each α-SNAP−SNARE subcomplex was treated as a rigid body (for a 20S particle with four engaged N-domains, this would be 17 total rigid body groups). Nucleotides, Mg²⁺, Pᵢ, and water molecules were then added where appropriate. Subsequent cycles of manual model building were performed in *Coot* and *ISOLDE* followed by 1–7 macrocycles of *phenix.real_space_refine* with global minimization and local grid search enabled. Ramachandran and secondary structure restraints were used throughout.

All model figures were prepared with *ChimeraX*, in many cases with custom CXC scripts generated by the Python scripts described below.

## Automated NSF model analysis and coordinate PCA

Several custom Python scripts (*preprocess_models.py, analyze_models.py*) using the *Bio.PDB* module[90] of the Biopython project[91] as well as *pandas*[71], *numpy*[72], *scipy*[73], *scikit-learn*[92], and *matplotlib*[74] were used to standardize all 25 NSF, sx20S, or bin20S models for further analysis, such as to perform principle components analysis (PCA) and HDBSCAN clustering, to measure NSF D1 large-small subdomain angles, to perform planarity fits for NSF domains and subdomains, to measure distances between key catalytic residues and nucleotides, and to map various values to structures for visualization in *UCSF ChimeraX*.

For PCA, 132 NSF D1 domains from 22 models over all datasets were split into large and small subdomains, which were then aligned to one another (large, residues 224–373; small, residues 400–486) using *Bio.PDB.Superimposer*. Splitting removes the largest source of variance in the data, the rigid body motions of the two subdomains relative to each other, which is considered trivial for the purposes of this analysis. Next, a Cα coordinate matrix **C** with dimensions $n_{models} \times n_{residues} \times 3$ was constructed for residues shared by all models and flattened into two dimensions, creating **C_flat** with $n_{models}$ rows and $n_{residues} \times 3$ columns. **C_flat** was then rescaled using *sklearn.preprocessing.StandardScaler* and PCA was performed on **C_flat** using *sklearn.decomposition.PCA*. 13 components were found to contribute 99% of the variance, so a transformed matrix **C_trans** with dimensions $n_{models} \times 13$, with eigenvectors sorted along the second axis by eigenvalue, was calculated. The cluster analysis of models was then performed using *sklearn.cluster.HDBSCAN* with a minimum cluster size of three, producing 11 clusters accounting for 119/132 total models. This analysis serves as the basis for Fig. 8A.

For NSF D1 and D2 plane fits, centers of mass for all subdomains were calculated, and the planes for different combinations of these points were fit using a least squares approach (Supplementary Fig. 8). For NSF D1 large-small subdomain angle calculations, the primary axes of two α-helices in the large (α3, residues 350–358) and small subdomains (α8, residues 442–452) were calculated by singular value decomposition using *np.linalg.svd*, with implementation inspired by *helix_analysis.py* from the *MDAnalysis* package[93]. Vectors corresponding to these helical axes $u$ and $v$ were normalized and centered at the coordinate origin, and the angle between them was calculated as

$$\theta = 2 \times \tan^{-1}\left( \sqrt{\sum_i \vec{u}_i^2} \sqrt{\sum_i \vec{v}_i^2} \right).$$

Distances between key active site atoms over all models (Fig. 9) were calculated using *np.linalg.norm*, where the norm of the difference vector of the coordinates of each atom is the distance between them in Å. Significant changes in protomer E active site residues relative to the other five protomers were assessed by pairwise application of the Welch's *t*-test assuming non-equal variances as implemented in *scipy.stats.ttest_ind*. Distance distributions were considered statistically different if $p < 0.05$.

All these values, as well as manually curated, qualitative data labels, were plotted in various ways using *matplotlib* or incorporated into automatically generated *ChimeraX* CXC scripts.

## Statistics and reproducibility

No statistical method was used to predetermine the sample size, but experiments described in this study were performed with at least three samples for each group in the experiments shown throughout.

## Reporting summary

Further information on research design is available in the Nature Portfolio Reporting Summary linked to this article.

## Data availability

The atomic coordinates have been deposited in the Protein Data Bank (PDB) under accession codes EMDB70529 and 9OJ2 (sx20S complex (NSF-alphaSNAP-syntaxin-1a), non-hydrolyzing, class 1); EMD-70536

and 9OJJ (sx20S complex (NSF-alphaSNAP-syntaxin-1a), alphaSNAP-syntaxin-1a subcomplex local refinement, non-hydrolyzing, class 1); EMDB-70547 and 9OJR (21bin20S complex (NSF-alphaSNAP-2:1 syntaxin-1a:SNAP-25), non-hydrolyzing, class 3); EMDB-70548 [] and 9OJU (21bin20S complex (NSF-alphaSNAP-2:1 syntaxin-1a:SNAP-25), non-hydrolyzing, class 4); EMDB-70550 and 9OJZ (21bin20S complex (NSF-alphaSNAP-2:1 syntaxin-1a:SNAP-25), non-hydrolyzing, class 5); EMDB-70553 and 9OK3 (21bin20S complex (NSF-alphaSNAP-2:1 syntaxin-1a:SNAP-25), 3:2:1 alphaSNAP-syntaxin-1a-SNAP-25 subcomplex local refinement, non-hydrolyzing, class 13); EMDB-70554 and 9OK5 (22bin20S complex (NSF-alphaSNAP-2:2 syntaxin-1a:SNAP-25), hydrolyzing, class 16); EMDB-70559 and 9OKC (22bin20S complex (NSF-alphaSNAP-2:2 syntaxin-1a:SNAP-25), hydrolyzing, class 17); EMDB-70594 and 9OLJ (22bin20S complex (NSF-alphaSNAP-2:2 syntaxin-1a:SNAP-25), hydrolyzing, class 18); EMDB-70598 and 9OLO (22bin20S complex (NSF-alphaSNAP-2:2 syntaxin-1a:SNAP-25), hydrolyzing, class 19); EMDB-70608 and 9OM6 (22bin20S complex (NSF-alphaSNAP-2:2 syntaxin-1a:SNAP-25), 4:2:2 alphaSNAP-syntaxin-1a-SNAP-25 subcomplex local refinement, hydrolyzing, class 23); EMDB-70616 and 9OMQ (NSF, substrate free, hydrolyzing, class 24); EMDB-71437 and 9PAF (21bin20S complex (NSF-alphaSNAP-2:1 syntaxin-1a:SNAP-25), non-hydrolyzing, class 6); EMDB-71438 and 9PAG (21bin20S complex (NSF-alphaSNAP-2:1 syntaxin-1a:SNAP-25), non-hydrolyzing, class 7); EMDB-71458 and 9PB9 (21bin20S complex (NSF-alphaSNAP-2:1 syntaxin−1a:SNAP-25), non-hydrolyzing, class 8); EMDB-71475 and 9PBA (21bin20S complex (NSF-alphaSNAP-2:1 syntaxin-1a:SNAP-25), non-hydrolyzing, class 9); EMDB-71478 and 9PBF (21bin20S complex (NSF-alphaSNAP-2:1 syntaxin-1a:SNAP-25), non-hydrolyzing, class 10); EMDB-71491 and 9PBV (21bin20S complex (NSF-alphaSNAP-2:1 syntaxin-1a:SNAP-25), non-hydrolyzing, class 11); EMDB-71496 and 9PC3 (21bin20S complex (NSF-alphaSNAP-2:1 syntaxin-1a:SNAP-25), non-hydrolyzing, class 12); EMDB-71521 and 9PCX (22bin20S complex (NSF-alphaSNAP-2:2 syntaxin-1a:SNAP-25), hydrolyzing, class 14); EMDB-71522 and 9PCZ (22bin20S complex (NSF-alphaSNAP-2:2 syntaxin-1a:SNAP-25), hydrolyzing, class 15); EMDB-71523 and 9PD1 (22bin20S complex (NSF-alphaSNAP-2:2 syntaxin-1a:SNAP-25), hydrolyzing, class 20); EMDB-71529 and 9PD8 (22bin20S complex (NSF-alphaSNAP-2:2 syntaxin-1a:SNAP-25), hydrolyzing, class 21); EMDB-71530 and 9PDB (22bin20S complex (NSF-alphaSNAP-2:2 syntaxin-1a:SNAP-25), hydrolyzing, class 22); EMDB-71533 and 9PDD (22bin20S complex (NSF-alphaSNAP-2:2 syntaxin-1a:SNAP-25), hydrolyzing, class 29); EMDB-71591 and 9PF2 (NSF, substrate free, hydrolyzing, class 25); EMDB-71598 and 9PFC (NSF, substrate free, hydrolyzing, class 26); EMDB-71600 and 9PFF (min22bin20S complex (NSF-alphaSNAP-2:2 syntaxin-1a H3:SNAP-25 SN1), non-hydrolyzing, class 27); and EMDB-71601 and 9PFG (min22bin20S complex (NSF-alphaSNAP-2:2 syntaxin-1a H3:SNAP-25 SN1), 4:2:2 alphaSNAP-syntaxin-1a H3-SNAP-25 SN1 subcomplex local refinement, non-hydrolyzing, class 28). Previously published models used in this work include 6MDM (The 20S supercomplex engaging the SNAP-25 N-terminus (class 1)); 1N7S (High Resolution Structure of a Truncated Neuronal SNARE Complex); 1QCS (N-TERMINAL DOMAIN OF N-ETHYLMALEIMIDE SENSITIVE FACTOR (NSF)); 1JTH (Crystal structure and biophysical properties of a complex between the N-terminal region of SNAP25 and the SNARE region of syntaxin 1a); and 1HVV (SELF-ASSOCIATION OF THE H3 REGION OF SYNTAXIN 1 A). A complete deposition including all models, code, and analysis reported in this work can be found in a corresponding Zenodo repository [https://doi.org/10.5281/zenodo.16422563] (Structural remodeling of target-SNARE protein complexes by NSF enables synaptic transmission: computational workflow and structural analysis).

## Code availability

We used *AlphaFold2* [94], *ChimeraX* 1.6–1.8 [84], *Coot* 0.9 [83], *CryoSPARC* [57,80], *CTFFIND4* [77], *ImageJ* [67], *ISOLDE* [85], *MolProbity* [95], *MotionCorr2* [76], *PHENIX* 1.2.X [86,87,96], *Prism* (GraphPad Software), *RELION* 3 [78], *Segger* [81], and *SerialEM* 4 [75]. Custom scripts were written in *Python* 3.1 with packages including *numpy* [72], *scipy* [73], *scikit-learn* [92], *biopython* [90,91], *pandas* [71], and *matplotlib* [74] and are available in the Zenodo deposition for this manuscript, [https://doi.org/10.5281/zenodo.16422563] (Structural remodeling of target-SNARE protein complexes by NSF enables synaptic transmission: computational workflow and structural analysis).

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

## Acknowledgements

We thank Richard Held, Jeremy Leitz, Rama Ranganathan, Subu Subramanian, Benjamin White, Susan Wu, and Minglei Zhao for stimulating discussions; E. Montabana and C. Zhang for assistance with the electron microscopy data collection at the Stanford University cEMc; and Paul D. Adams, Pavel Afonine, and Nigel Moriarty for assistance with PHENIX refinement. Funding was provided by the US National Institutes of Health (NIH; RO1MH63105 to A.T.B.) and the Helen Hay Whitney Foundation/HHMI (to K.I.W.). Molecular graphics and analyses performed with *UCSF ChimeraX*, developed by the Resource for Biocomputing, Visualization, and Informatics at the University of California, San Francisco, with support from National Institutes of Health R01-GM129325 and the Office of Cyber Infrastructure and Computational Biology, National Institute of Allergy and Infectious Diseases. This article is subject to HHMI's Open Access to Publications policy. HHMI laboratory heads have previously granted a non-exclusive CC BY 4.0 license to the public and a sublicensable license to HHMI in their research articles. Pursuant to those licenses, the author-accepted manuscript of this article can be made freely available under a CC BY 4.0 license immediately upon publication.

## Author contributions

K.I.W. and A.T.B. developed the research plan and experimental strategy. K.Q. and J.D. performed the cellular imaging experiments in PC12 cells and analyzed the data. Y.A.K. and A.B. performed the cellular imaging experiments in yeast and analyzed the data. K.I.W., Y.A.K., S.C., L.E., and R.P. purified all proteins. K.I.W., S.C., and Y.A.K. performed the disassembly experiments. K.I.W. performed the cryo-EM experiments and analyzed the data. K.I.W. and A.T.B. wrote the manuscript with input from the other authors.

## Competing interests

The authors declare no competing interests.
