## [Transparent Peer Review file · Nature Communications]

Structural remodeling of target-SNARE protein complexes by NSF enables synaptic transmission

Corresponding Author: Dr Axel Brunger

Version 0:

Reviewer comments:

Reviewer #1

(Remarks to the Author)

This is an interesting new study by White et al, examining NSF/SNAP disassembly of SNARE complexes that are non-canonical, i.e. SNARE complexes that contain non-fuseogenic combinations of multiple syntaxins and SNAP-25. These combinations may possibly form at the synapse, and if so, would require disassembly by NSF/SNAP and ATP hydrolysis, and promote binding of Munc18. Here the authors show that disassembly of these complexes in vitro can occur, further indicating the flexibility of NSF/SNAP to bind and disassemble similar four-helical-bundle structures. In determining and analyzing many new structures and structural classes, the authors gained new insights into sequential hydrolysis by NSF and allosteric regulation. The authors conclude that cellular clusters of syntaxin-1a may serve as reservoirs that could be disassembled by NSF/SNAP for subsequent binding of Munc18, followed by SNARE complex assembly and vesicle fusion. The authors have done a reasonable job responding to the reviewers' concerns, including a new Fig 1E panel demonstrating disassembly of a tetrameric syntaxin-1a complex and binding of Munc18. Although I still mildly disagree on several points raised, I think that this is strong body of work, and the conclusions proposed by the authors are totally satisfactory for publication at this point.

Reviewer #2

(Remarks to the Author)

I agree with the overall tenor of Reviewer 1's comments. Specifically, (1) the overall quality of this work is exceptionally high; (2) the findings and analysis will greatly interest the AAA+ community; and (3) the focus on more speculative synaptic biology can be distracting (e.g., the first five paragraphs of the Discussion are given over to this topic!). But this is ultimately up to the authors. Overall, the strengths greatly outweigh the weaknesses, at least for this reviewer, and I recommend expeditious publication in Nature Communications.

1. Do the authors mean to imply that antiparallel SNARE complexes are not substrates for NSF/SNAP? And if so, would this mean that such complexes don't form in vivo, or that they are disassembled in some other way?
2. Line 61: The Habc domain is N-terminal, not C-terminal.
3. Fig. S1A shows very little 20S complex. First, how do the authors understand the low yield, and what might it mean? Second, is the low yield consistent with the claim that 5% of the particles had bound SNAREs?
4. Line 326-7: "(Fig 5A, left, top, and bottom, respectively...)" should be bottom and top respectively.
5. Lines 347-9: The sentence beginning "Furthermore, classes..." should be rewritten for clarity. For what it's worth, I preferred the explanation in the legend to Fig. S5b.
6. Four figure legends refer to "dashed shading [or lines] and an asterisk". The wisdom of this notation was lost on me; the authors should reconsider.

7. Fig. 5A: In the right-hand panel, there is an orange object that seems to be vestigial.

8. In a manuscript of this complexity, it is perhaps unsurprising that the figures and their textual citations have gotten a bit messed up. There are citations to figures that don't exist (e.g., S11-13, 12C-D, S9B, 7B) or are incorrect (S4-5 instead of Extended Figs. 4-5). Figures are cited out of order (e.g., S2 after S6). I'm sure there are other examples, which the authors should endeavor to discover and correct.

Response to Reviewers

The author's comments are in black, and our responses are in blue. We have also uploaded a manuscript file with significant changes highlighted.

Reviewer #1 (Comments for the Author):

This is an interesting new study by White et al, examining NSF/SNAP disassembly of SNARE complexes that are non-canonical, i.e. SNARE complexes that contain non-fuseogenic combinations of multiple syntaxins and SNAP-25. These combinations may possibly form at the synapse, and if so, would require disassembly by NSF/SNAP and ATP hydrolysis, and promote binding of Munc18. Here the authors show that disassembly of these complexes *in vitro* can occur, further indicating the flexibility of NSF/SNAP to bind and disassemble similar four-helical-bundle structures. In determining and analyzing many new structures and structural classes, the authors gained new insights into sequential hydrolysis by NSF and allosteric regulation. The authors conclude that cellular clusters of syntaxin-1a may serve as reservoirs that could be disassembled by NSF/SNAP for subsequent binding of Munc18, followed by SNARE complex assembly and vesicle fusion. The authors have done a reasonable job responding to the reviewers' concerns, including a new Fig 1E panel demonstrating disassembly of a tetrameric syntaxin-1a complex and binding of Munc18. Although I still mildly disagree on several points raised, I think that this is strong body of work, and the conclusions proposed by the authors are totally satisfactory for publication at this point.

We thank the reviewer for their kind words and are glad to hear that our previous response was satisfactory.

Reviewer #2 (Remarks to the Author):

I agree with the overall tenor of Reviewer 1's comments. Specifically, (1) the overall quality of this work is exceptionally high; (2) the findings and analysis will greatly interest the AAA+ community; and (3) the focus on more speculative synaptic biology can be distracting (e.g., the first five paragraphs of the Discussion are given over to this topic!). But this is ultimately up to the authors. Overall, the strengths greatly outweigh the weaknesses, at least for this reviewer, and I recommend expeditious publication in Nature Communications.

We thank the reviewer for their kind words.

1. Do the authors mean to imply that antiparallel SNARE complexes are not substrates for NSF/SNAP? And if so, would this mean that such complexes don't form *in vivo*, or that they are disassembled in some other way?

We thank the reviewer for this fascinating question.

The formation of anti-parallel SNARE complexes from soluble constructs *in vitro* is well known (e.g., refs. 1, 2), and anti-parallel SNARE complexes are rapidly disassembled in the presence of alpha-SNAP and NSF (ref. 2). However, all EM studies of the 20S complex to date, from our lab (2, 3, 4, 5) and others (6), reveal the presence of parallel, but not anti-parallel, SNARE complexes engaged by alpha-SNAP and NSF. This may be a consequence of the specific nature of alpha-SNAP recognition of the parallel SNARE bundle; a typical parallel SNARE complex has a uniform, left-handed twist from the N- to C-terminal ends of the bundle, which is complemented by at least two alpha-SNAPs binding adjacent to one another with a right-handed twist. While the crystal structure of a tetrameric syntaxin H3 complex composed of an antiparallel dimer of dimers roughly maintains a left-handed twist, the overall shape of the bundle is different and less uniform (ref. 7); it is unclear to what degree the alpha-SNAP—SNARE interactions reported here or elsewhere (ref. 6) would be maintained. In short, anti-parallel species are disassembled *in vitro* by NSF and alpha-SNAP, but they may be too rare or heterogeneous to be observable by cryo-EM.

More generally, we note that anti-parallel SNARE complexes have thus far only been observed *in vitro* with soluble SNARE proteins, but this does not preclude the possibility of the formation or disassembly of anti-parallel complexes between membrane-anchored SNARE proteins in the cell by NSF and alpha-SNAP.

We have revised the text accordingly.

2. Line 61: The Habc domain is N-terminal, not C-terminal.

Good catch! We have fixed this.

3. Fig. S1A shows very little 20S complex. First, how do the authors understand the low yield, and what might it mean? Second, is the low yield consistent with the claim that 5% of the particles had bound SNAREs?

The cryo-EM experiment described in this manuscript was performed before it was clear what to expect (i.e., would NSF engage any syntaxin oligomer at all?), so we chose a syntaxin concentration for sx20S complex formation based on a simplifying assumption of monomeric syntaxin in solution, and mixed NSF:syntaxin:alpha-SNAP with the standard ratio of 1:5:30 as in the other experiments. Given the starting concentration of NSF in these experiments (10 μM), this led to a hypothetical monomeric syntaxin concentration of 50 μM , a bit close in hindsight to the reported oligomerization threshold determined previously by CD spectroscopy (ref. 8, Fig. 5, reproduced below). This led to a low yield of the sx20S complex, with around 5% of NSF forming sx20S complex with syntaxin and alpha-SNAP in the EM experiment, as the reviewer notes. This demonstrates that the effective concentration of tetrameric syntaxin in the original prep was probably quite low. This result is also consistent with the detailed mass spectroscopy observations recently published by Hesselbarth and Schmidt (1). In summary, the low yield makes sense if we assume that our preps of soluble syntaxin are characterized by a concentration-dependent mixture of monomeric and oligomeric species, and that NSF and alpha-SNAP mostly form stable complexes with parallel, tetrameric oligomers (related to question 1 above).

Based on this reasoning, we subsequently increased the syntaxin concentration in our assays for tetramer disassembly and Munc18 capture by tenfold to ensure that we were at concentrations exceeding the threshold for full oligomerization and to ensure that the absolute concentration of parallel tetramer was sufficiently high. This difference is evident upon comparison of Supplementary Fig. 1A (complex prep for cryo-EM) and Ext. Data Fig. 2A (complex prep for disassembly/M18 capture).

Conceptually, we note that *in vivo*, topography and the chemical environment of the membrane likely enrich for the formation of parallel, oligomeric species over the levels seen *in vitro* with soluble syntaxin. In particular, the TMR of syntaxin ensures that it cannot fully sample rotational space, promoting parallel SNARE-SNARE interactions at the expense of antiparallel oligomeric syntaxin. Clustering related to interactions with membrane lipids or other factors would also enhance the effective local concentration of syntaxin and would act to drive it into the oligomeric state.

We have added this discussion to the SI and refer the reader to it in the main text.

FIG. 5. **Progressive decrease in α -helicity upon dilution of syntaxin SNARE motifs.** CD spectra were recorded at three different protein concentrations (40 mM sodium phosphate buffer). Note that in this experiment a slightly longer fragment of syntaxin was used (residues 180–262 instead of 183–262).

4. Line 326-7: “(Fig 5A, left, top, and bottom, respectively...)” should be bottom and top respectively.

Thank you for catching this error. We have fixed this.

5. Lines 347-9: The sentence beginning “Furthermore, classes...” should be rewritten for clarity. For what it’s worth, I preferred the explanation in the legend to Fig. S5b.

Thank you—we have revised this sentence for clarity.

6. Four figure legends refer to “dashed shading [or lines] and an asterisk”. The wisdom of this notation was lost on me; the authors should reconsider.

Thank you for the feedback—we have simplified the description for clarity.

7. Fig. 5A: In the right-hand panel, there is an orange object that seems to be vestigial.

We assume the reviewer is referring to the orange protein model that partially occupies the split from the axial perspective in Fig. 5A—this is the ATP-bound D2 domain of protomer A. We applied a clipping plane to create these images, rather than completely hiding maps and models. This helps to illustrate the large degree of conformational change that accompanies the splitting of D1/D2 in the post-hydrolysis state. We recognize that this may be confusing and have made explicit mention of this in the figure legend.

8. In a manuscript of this complexity, it is perhaps unsurprising that the figures and their textual citations have gotten a bit messed up. There are citations to figures that don’t exist (e.g., S11-13, 12C-D, S9B, 7B) or are incorrect (S4-5 instead of Extended Figs. 4-5). Figures are cited out of order (e.g., S2 after S6). I’m sure there are other examples, which the authors should endeavor to discover and correct.

Thank you for catching these errors—we have now fixed them.

In addition, we have rearranged the following figures to better match the order in which they are referenced in the text:

- Ext. Data Fig. 7 to 8
- Ext. Data Fig. 8 to 7
- Supp. Fig. 2 to 5
- Supp. Fig. 3 to 7
- Supp. Fig. 4 to 2
- Supp. Fig. 5 to 3
- Supp. Fig. 6 to 4
- Supp. Fig. 7 to 6

We have also rearranged the order of the panels in Ext. Data Fig. 2 to better match the order in which they are referenced in text.

9.) Please see attached document for comments on PDB validation reports

We thank the reviewer for reviewing the validation reports in detail. We are currently addressing outstanding issues with the depositions, but it has been a slow process due to delays on the RCSB side.

First, we note that many of the comments related to which residues were modeled in the maps, and the quality of some of the fits. The sequences in the PDB files reflect the constructs that were used. However, in multiple cases, only a fraction of some of the chains were modeled as the density maps were poor in corresponding regions. In the case of NSF, we omitted the N-domains where the density was too degenerate to resolve the lobed shape of the domain, as rigid body fits of the crystal structure into the density envelope were not reliable. In the case of the 2:2 binary SNARE complexes, entire SNAP-25 linkers and SN2 domains are unresolved, leading to more than half of the protein appearing unmodeled in the validation report. These regions of the protein extend away from the 20S complex, beyond the spire and into solvent, where they likely sample a huge conformational space. Similarly, the Habc domains of syntaxin molecules could not be resolved.

Second, we chose to dock models derived from local refinements of the alpha-SNAP/SNARE subcomplexes back into the full maps, as they are consistent with the overall spire density. Typically, the spire density produced from the global refinements is sufficient to resolve alpha-SNAP/SNARE secondary structure needed to place the subcomplex, but side chain density is not present. As a result, the map-model correlation between the spire model components and the full map is weak in this region. We chose this strategy rather than creating composite maps—the lesser of two evils, in our opinion. We have expanded the description of our modeling choices in the Methods section (see Model Building and Refinement) to clarify these points and will add comments to the PDB depositions.

Third, with regards to a few of the sequence conflicts with respect to P60881 (NSF), we note that most of these are indeed related to expression tags, however in a few of the models residues 485–487 are modeled as AsnAspGlu but should instead be modeled as GluAsnAsp. We are correcting this issue now for our final deposition. Given the fact that these surface exposed residues are of similar size and lead into the D1-D2 linker, fixing this will lead to very minimal changes to the models. Finally, we will also address the remaining close clashes in the models.

References

1. Hesselbarth, J., and Schmidt, C. “Mass Spectrometry Uncovers Intermediates and Off-Pathway Complexes for SNARE Complex Assembly.” *Communications Biology* 6, no. 1 (February 20, 2023): 1–15. <https://doi.org/10.1038/s42003-023-04548-0>.
2. Choi, U.B., Zhao, M., White, K.I., Pfuetzner, R.A., Esquivias, L., Zhou, Q., Brunger, A.T. (2018). NSF-mediated disassembly of on and off-pathway SNARE complexes and inhibition by complexin. *eLife* 7, e36497, 10.7554/eLife.36497 (2018).
3. Zhao, Minglei, Shenping Wu, Qiangjun Zhou, Sandro Vivona, Daniel J. Cipriano, Yifan Cheng, and Axel T. Brunger. “Mechanistic Insights into the Recycling Machine of the SNARE Complex.” *Nature* 518, no. 7537 (February 2015): 61–67. <https://doi.org/10.1038/nature14148>.
4. White, K.I., Zhao, M., Choi, U.B., Pfuetzner, R.A., and Brunger, A.T. “Structural Principles of SNARE Complex Recognition by the AAA+ Protein NSF.” *eLife* 7 (September 10, 2018): e38888. <https://doi.org/10.7554/eLife.38888>.
5. “Sec18 Side-Loading Is Essential for Universal SNARE Recycling across Cellular Contexts | bioRxiv.” Accessed May 30, 2025. <https://www.biorxiv.org/content/10.1101/2024.08.30.610324v1>. Accepted, NSMB.
6. Huang, X., Sun, S., Wang X., Fan, F., Zhou, Q., Lu, S., Cao, Y., et al. “Mechanistic Insights into the SNARE Complex Disassembly.” *Science Advances* 5, no. 4 (April 10, 2019): eaau8164. <https://doi.org/10.1126/sciadv.aau8164>.
7. Misura, Kira M. S., Richard H. Scheller, and William I. Weis. “Self-Association of the H3 Region of Syntaxin 1A.” *Journal of Biological Chemistry* 276, no. 16 (April 20, 2001): 13273–82. <https://doi.org/10.1074/jbc.M009636200>.
8. Margittai, Martin, Dirk Fasshauer, Stefan Pabst, Reinhard Jahn, and Ralf Langen. “Homo- and Heterooligomeric SNARE Complexes Studied by Site-Directed Spin Labeling *.” *Journal of Biological Chemistry* 276, no. 16 (April 20, 2001): 13169–77. <https://doi.org/10.1074/jbc.M010653200>.

Response to Reviewers

The author's comments are in black, and our responses are in blue. We have also uploaded a manuscript file with significant changes highlighted.

Reviewer #1 (Comments for the Author):

This is an interesting new study by White et al, examining NSF/SNAP disassembly of SNARE complexes that are non-canonical, i.e. SNARE complexes that contain non-fuseogenic combinations of multiple syntaxins and SNAP-25. These combinations may possibly form at the synapse, and if so, would require disassembly by NSF/SNAP and ATP hydrolysis, and promote binding of Munc18. Here the authors show that disassembly of these complexes *in vitro* can occur, further indicating the flexibility of NSF/SNAP to bind and disassemble similar four-helical-bundle structures. In determining and analyzing many new structures and structural classes, the authors gained new insights into sequential hydrolysis by NSF and allosteric regulation. The authors conclude that cellular clusters of syntaxin-1a may serve as reservoirs that could be disassembled by NSF/SNAP for subsequent binding of Munc18, followed by SNARE complex assembly and vesicle fusion. The authors have done a reasonable job responding to the reviewers' concerns, including a new Fig 1E panel demonstrating disassembly of a tetrameric syntaxin-1a complex and binding of Munc18. Although I still mildly disagree on several points raised, I think that this is strong body of work, and the conclusions proposed by the authors are totally satisfactory for publication at this point.

We thank the reviewer for their kind words and are glad to hear that our previous response was satisfactory.

Reviewer #2 (Remarks to the Author):

I agree with the overall tenor of Reviewer 1's comments. Specifically, (1) the overall quality of this work is exceptionally high; (2) the findings and analysis will greatly interest the AAA+ community; and (3) the focus on more speculative synaptic biology can be distracting (e.g., the first five paragraphs of the Discussion are given over to this topic!). But this is ultimately up to the authors. Overall, the strengths greatly outweigh the weaknesses, at least for this reviewer, and I recommend expeditious publication in Nature Communications.

We thank the reviewer for their kind words.

1. Do the authors mean to imply that antiparallel SNARE complexes are not substrates for NSF/SNAP? And if so, would this mean that such complexes don't form *in vivo*, or that they are disassembled in some other way?

We thank the reviewer for this fascinating question.

The formation of anti-parallel SNARE complexes from soluble constructs *in vitro* is well known (e.g., refs. 1, 2), and anti-parallel SNARE complexes are rapidly disassembled in the presence of alpha-SNAP and NSF (ref. 2). However, all EM studies of the 20S complex to date, from our lab (2, 3, 4, 5) and others (6), reveal the presence of parallel, but not anti-parallel, SNARE complexes engaged by alpha-SNAP and NSF. This may be a consequence of the specific nature of alpha-SNAP recognition of the parallel SNARE bundle; a typical parallel SNARE complex has a uniform, left-handed twist from the N- to C-terminal ends of the bundle, which is complemented by at least two alpha-SNAPs binding adjacent to one another with a right-handed twist. While the crystal structure of a tetrameric syntaxin H3 complex composed of an antiparallel dimer of dimers roughly maintains a left-handed twist, the overall shape of the bundle is different and less uniform (ref. 7); it is unclear to what degree the alpha-SNAP—SNARE interactions reported here or elsewhere (ref. 6) would be maintained. In short, anti-parallel species are disassembled *in vitro* by NSF and alpha-SNAP, but they may be too rare or heterogeneous to be observable by cryo-EM.

More generally, we note that anti-parallel SNARE complexes have thus far only been observed *in vitro* with soluble SNARE proteins, but this does not preclude the possibility of the formation or disassembly of anti-parallel complexes between membrane-anchored SNARE proteins in the cell by NSF and alpha-SNAP.

We have revised the text accordingly.

2. Line 61: The Habc domain is N-terminal, not C-terminal.

Good catch! We have fixed this.

3. Fig. S1A shows very little 20S complex. First, how do the authors understand the low yield, and what might it mean? Second, is the low yield consistent with the claim that 5% of the particles had bound SNAREs?

The cryo-EM experiment described in this manuscript was performed before it was clear what to expect (i.e., would NSF engage any syntaxin oligomer at all?), so we chose a syntaxin concentration for sx20S complex formation based on a simplifying assumption of monomeric syntaxin in solution, and mixed NSF:syntaxin:alpha-SNAP with the standard ratio of 1:5:30 as in the other experiments. Given the starting concentration of NSF in these experiments (10 μM), this led to a hypothetical monomeric syntaxin concentration of 50 μM , a bit close in hindsight to the reported oligomerization threshold determined previously by CD spectroscopy (ref. 8, Fig. 5, reproduced below). This led to a low yield of the sx20S complex, with around 5% of NSF forming sx20S complex with syntaxin and alpha-SNAP in the EM experiment, as the reviewer notes. This demonstrates that the effective concentration of tetrameric syntaxin in the original prep was probably quite low. This result is also consistent with the detailed mass spectroscopy observations recently published by Hesselbarth and Schmidt (1). In summary, the low yield makes sense if we assume that our preps of soluble syntaxin are characterized by a concentration-dependent mixture of monomeric and oligomeric species, and that NSF and alpha-SNAP mostly form stable complexes with parallel, tetrameric oligomers (related to question 1 above).

Based on this reasoning, we subsequently increased the syntaxin concentration in our assays for tetramer disassembly and Munc18 capture by tenfold to ensure that we were at concentrations exceeding the threshold for full oligomerization and to ensure that the absolute concentration of parallel tetramer was sufficiently high. This difference is evident upon comparison of Supplementary Fig. 1A (complex prep for cryo-EM) and Ext. Data Fig. 2A (complex prep for disassembly/M18 capture).

Conceptually, we note that *in vivo*, topography and the chemical environment of the membrane likely enrich for the formation of parallel, oligomeric species over the levels seen *in vitro* with soluble syntaxin. In particular, the TMR of syntaxin ensures that it cannot fully sample rotational space, promoting parallel SNARE-SNARE interactions at the expense of antiparallel oligomeric syntaxin. Clustering related to interactions with membrane lipids or other factors would also enhance the effective local concentration of syntaxin and would act to drive it into the oligomeric state.

We have added this discussion to the SI and refer the reader to it in the main text.

FIG. 5. Progressive decrease in α -helicity upon dilution of syntaxin SNARE motifs. CD spectra were recorded at three different protein concentrations (40 mM sodium phosphate buffer). Note that in this experiment a slightly longer fragment of syntaxin was used (residues 180–262 instead of 183–262).

4. Line 326-7: “(Fig 5A, left, top, and bottom, respectively...)” should be bottom and top respectively.

Thank you for catching this error. We have fixed this.

5. Lines 347-9: The sentence beginning “Furthermore, classes...” should be rewritten for clarity. For what it’s worth, I preferred the explanation in the legend to Fig. S5b.

Thank you—we have revised this sentence for clarity.

6. Four figure legends refer to “dashed shading [or lines] and an asterisk”. The wisdom of this notation was lost on me; the authors should reconsider.

Thank you for the feedback—we have simplified the description for clarity.

7. Fig. 5A: In the right-hand panel, there is an orange object that seems to be vestigial.

We assume the reviewer is referring to the orange protein model that partially occupies the split from the axial perspective in Fig. 5A—this is the ATP-bound D2 domain of protomer A. We applied a clipping plane to create these images, rather than completely hiding maps and models. This helps to illustrate the large degree of conformational change that accompanies the splitting of D1/D2 in the post-hydrolysis state. We recognize that this may be confusing and have made explicit mention of this in the figure legend.

8. In a manuscript of this complexity, it is perhaps unsurprising that the figures and their textual citations have gotten a bit messed up. There are citations to figures that don't exist (e.g., S11-13, 12C-D, S9B, 7B) or are incorrect (S4-5 instead of Extended Figs. 4-5). Figures are cited out of order (e.g., S2 after S6). I'm sure there are other examples, which the authors should endeavor to discover and correct.

Thank you for catching these errors—we have now fixed them.

In addition, we have rearranged the following figures to better match the order in which they are referenced in the text:

- Ext. Data Fig. 7 to 8
- Ext. Data Fig. 8 to 7
- Supp. Fig. 2 to 5
- Supp. Fig. 3 to 7
- Supp. Fig. 4 to 2
- Supp. Fig. 5 to 3
- Supp. Fig. 6 to 4
- Supp. Fig. 7 to 6

We have also rearranged the order of the panels in Ext. Data Fig. 2 to better match the order in which they are referenced in text.

9.) Please see attached document for comments on PDB validation reports

We thank the reviewer for reviewing the validation reports in detail. We are currently addressing outstanding issues with the depositions, but it has been a slow process due to delays on the RCSB side.

First, we note that many of the comments related to which residues were modeled in the maps, and the quality of some of the fits. The sequences in the PDB files reflect the constructs that were used. However, in multiple cases, only a fraction of some of the chains were modeled as the density maps were poor in corresponding regions. In the case of NSF, we omitted the N-domains where the density was too degenerate to resolve the lobed shape of the domain, as rigid body fits of the crystal structure into the density envelope were not reliable. In the case of the 2:2 binary SNARE complexes, entire SNAP-25 linkers and SN2 domains are unresolved, leading to more than half of the protein appearing unmodeled in the validation report. These regions of the protein extend away from the 20S complex, beyond the spire and into solvent, where they likely sample a huge conformational space. Similarly, the Habc domains of syntaxin molecules could not be resolved.

Second, we chose to dock models derived from local refinements of the alpha-SNAP/SNARE subcomplexes back into the full maps, as they are consistent with the overall spire density. Typically, the spire density produced from the global refinements is sufficient to resolve alpha-SNAP/SNARE secondary structure needed to place the subcomplex, but side chain density is not present. As a result, the map-model correlation between the spire model components and the full map is weak in this region. We chose this strategy rather than creating composite maps—the lesser of two evils, in our opinion. We have expanded the description of our modeling choices in the Methods section (see Model Building and Refinement) to clarify these points and will add comments to the PDB depositions.

Third, with regards to a few of the sequence conflicts with respect to P60881 (NSF), we note that most of these are indeed related to expression tags, however in a few of the models residues 485–487 are modeled as AsnAspGlu but should instead be modeled as GluAsnAsp. We are correcting this issue now for our final deposition. Given the fact that these surface exposed residues are of similar size and lead into the D1-D2 linker, fixing this will lead to very minimal changes to the models. Finally, we will also address the remaining close clashes in the models.

References

1. Hesselbarth, J., and Schmidt, C. "Mass Spectrometry Uncovers Intermediates and Off-Pathway Complexes for SNARE Complex Assembly." *Communications Biology* 6, no. 1 (February 20, 2023): 1–15. <https://doi.org/10.1038/s42003-023-04548-0>.
2. Choi, U.B., Zhao, M., White, K.I., Pfuetzner, R.A., Esquivies, L., Zhou, Q., Brunger, A.T. (2018). NSF-mediated disassembly of on and off-pathway SNARE complexes and inhibition by complexin. *eLife* 7, e36497, 10.7554/eLife.36497 (2018).
3. Zhao, Minglei, Shenping Wu, Qiangjun Zhou, Sandro Vivona, Daniel J. Cipriano, Yifan Cheng, and Axel T. Brunger. "Mechanistic Insights into the Recycling Machine of the SNARE Complex." *Nature* 518, no. 7537 (February 2015): 61–67. <https://doi.org/10.1038/nature14148>.
4. White, K.I., Zhao, M., Choi, U.B., Pfuetzner, R.A., and Brunger, A.T. "Structural Principles of SNARE Complex Recognition by the AAA+ Protein NSF." *eLife* 7 (September 10, 2018): e38888. <https://doi.org/10.7554/eLife.38888>.
5. "Sec18 Side-Loading Is Essential for Universal SNARE Recycling across Cellular Contexts | bioRxiv." Accessed May 30, 2025. <https://www.biorxiv.org/content/10.1101/2024.08.30.610324v1>. Accepted, NSMB.
6. Huang, X., Sun, S., Wang X., Fan, F., Zhou, Q., Lu, S., Cao, Y., et al. "Mechanistic Insights into the SNARE Complex Disassembly." *Science Advances* 5, no. 4 (April 10, 2019): eaau8164. <https://doi.org/10.1126/sciadv.aau8164>.
7. Misura, Kira M. S., Richard H. Scheller, and William I. Weis. "Self-Association of the H3 Region of Syntaxin 1A." *Journal of Biological Chemistry* 276, no. 16 (April 20, 2001): 13273–82. <https://doi.org/10.1074/jbc.M009636200>.
8. Margittai, Martin, Dirk Fasshauer, Stefan Pabst, Reinhard Jahn, and Ralf Langen. "Homo- and Heterooligomeric SNARE Complexes Studied by Site-Directed Spin Labeling *." *Journal of Biological Chemistry* 276, no. 16 (April 20, 2001): 13169–77. <https://doi.org/10.1074/jbc.M010653200>.